# Analysis of Honing Material Removal Rate and Surface Quality Using Electroplated Oilstone

**DOI:** 10.3390/ma17246170

**Published:** 2024-12-17

**Authors:** Hao Su, Changyong Yang, Yucan Fu, Rui Nie

**Affiliations:** 1Ningbo Institute of Technology, Beihang University, Ningbo 315800, China; 2Ningbo Key Laboratory for Advanced Delivery Equipment, Ningbo Institute of Technology, Ningbo 315800, China; 3College of Mechanical and Electrical Engineering, Nanjing University of Aeronautics and Astronautics, Nanjing 210016, China; yangchy@nuaa.edu.cn (C.Y.); yucanfu@nuaa.edu.cn (Y.F.)

**Keywords:** electro-hydraulic servo valve, honing, oilstone, processing efficiency, surface quality

## Abstract

The manufacturing precision of electro-hydraulic servo valve sleeves is critical to the performance and longevity of the valves. To ensure the service life of these valves, the valve sleeve is typically made from high-hardness martensitic stainless steel, which is considered a hard-to-cut material. Current honing methods often suffer from inefficiency and instability. This study compares the honing processes using electroplated and sintered oilstones, emphasizing processing efficiency and surface quality. Initially, the morphology of the oilstones was examined. Equivalent honing depth and material removal rate per unit width were developed to characterized material removal. The influence of various parameters on honing depth and material removal rates was explored, along with the surface morphology and roughness after honing. The results indicated that electroplated oilstones achieved a material removal rate 2.5 times higher than sintered oilstones. In contrast, sintered oilstones produced superior surface quality. To optimize both surface quality and efficiency, we proposed a sequential honing method: using electroplated oilstones for significant material removal followed by sintered oilstones for surface finishing, which enhanced efficiency by 1.6 times. Electroplated oilstone has broad application prospects in the field of precision and efficient machining of hydraulic components.

## 1. Introduction

Electro-hydraulic servo valves are essential components in closed-loop control systems, where precise adjustment of the orifice area is achieved through the relative motion between the valve spool and the valve sleeve, thereby accurately controlling the flow rate. However, the performance of these valves is highly sensitive to manufacturing errors in the valve spool and sleeve, including shape, position, and dimensional accuracy, and surface roughness [1].

For the valve sleeve, typical manufacturing accuracy requirements include a cylindricity error between 0.001 and 0.003 mm and a surface roughness of 0.1 μm. The fit between the valve sleeve and the valve spool usually maintains a minimal clearance in the micrometer range, which ensures that the valve spool can slide within the sleeve under gravity alone, without blockage.

The precise machining process of the electro-hydraulic servo valve sleeve primarily includes honing, heat treatment, and lapping. The final machining accuracy relies on manual lapping. Advanced honing machines feature multiple workstations and online measurement capabilities, allowing for rough and fine honing to be completed in a single clamping. However, operators still need to adjust the processing parameters based on the measurement results. Another lapping process uses the valve sleeve as a reference and employs a high-precision cylindrical grinder to achieve the appropriate valve spool assembly clearance, which ensures a consistent fit, but the corresponding valve sleeve still requires honing to achieve good shape accuracy. The honing process is widely used in machining hydraulic components such as valve sleeves, not only because of its advantages in shape accuracy but also due to the cross-hatch pattern it creates on the surface. This pattern is beneficial for storing lubricating oil, reducing friction between sliding valve components, and thereby extending the lifespan of the parts. Researches on honing surface texture mainly focus on the surface measurement of three-dimensional morphology and cross-hatch angle [2,3,4,5,6]. Beyerer used image processing algorithms to detect surface defects on the cylinder bore after honing processing, such as folded metal, groove interrupts, smudgy groove edges, etc. [2]. Jablonski digitally created a 3D model of a metal surface, which possessed characteristic features of a honed surface [3]. A new mapping method, combining SEM images and quantitative image analysis with traditional 2D profilometry, was developed for cylinder bores [4]. By extracting information from 2D data, honing surface features can be obtained, like the honing angle, groove parameters, surface defects, etc. [5]. The influence of beam angle in optical interferometry on the measurement of surface morphology parameters of honed cylinder holes was studied [6]. Moreover, the analysis of the cross-hatch textures resulted in many intersecting grooves on the engine cylinder block, which are conducive to the storage of oil, thereby improving the lubrication performance between the engine cylinder block and piston, reducing the wear of the cylinder bore, and thus increasing the life of the engine [7,8,9]. Spencer explored the effect of different honing angles on oil film thickness [7]. In particular, a flow continuity lubrication model considering the surface morphology of honing has been established, which has a guiding role in improving engine performance through machining [9]. Therefore, research on honing technology for electro-hydraulic servo valve sleeves is particularly important.

High-performance oilstones are critical to ensuring the quality and efficiency of the valve sleeve. During the honing of the servo valve sleeve, the grains in the oilstone continuously cut the workpiece, causing rapid wear of the grains. Additionally, the inner surface of the valve sleeve serves as a mating surface with stringent machining requirements. The accuracy of honing largely depends on the quality of the oilstone. The wear of the oilstone affects the machining quality, especially honing stability. Therefore, producing high-precision, wear-resistant oilstones is crucial for achieving high-efficiency, high-precision, and high-surface-quality machining of servo valve sleeves.

With the popularization of superabrasives, diamond and cubic boron nitride (cBN) abrasives have been widely used in the preparation of oilstones, greatly improving the wear resistance and extending their service life. Sabri studied the wear resistance and surface quality of workpieces machined with diamond oilstones featuring different binders [10,11]. Superabrasive oilstones have better surface processing quality than traditional abrasive oilstones; metal-bonded diamond oilstones have a longer life and are less prone to wear, but the surface quality for the same grit size and concentration is inferior to that of ceramic and resin-bonded diamond oilstones. Zhu honed martensitic stainless steel and titanium alloy materials with oilstones of different abrasives and concentrations, finding that various materials are best suited to specific types of oilstones [12]. Wang investigated the impact of different honing parameters on the wear performance of oilstones [13]. Cabanettes used oilstones at different wear stages to hone cylinder blocks, exploring the relationship between oilstone wear and surface roughness [14]. Furthermore, Yanshan University prepared oilstones with Cu-Sn binders, studying the effects of Sn content and alloy elements such as Fe, Co, and Ni on the binder structure and performance. Based on the experimental results, the binder composition was optimized to achieve good surface quality [15,16]. With the development of deep learning, neural networks and particle swarm algorithms were used to predict the wear of oilstones, facilitating reasonable oilstone replacement and maximizing oilstone usage while ensuring machining accuracy [17]. Thus, the introduction of superabrasives has significantly improved the wear resistance and service life of sintered oilstones, but issues with binder wear and chip accommodation space remain unresolved.

With the continuous development of honing technology, there is an increasing shift away from traditional sintered oilstones, and single-layer brazing technology has been applied to the honing tools [18]. Additionally, Kadia have replaced traditional oilstones with electroplated oilstones, achieving good processing results on high-speed honing machines. Researchers are exploring new oilstone preparation processes to achieve higher chip accommodation space, fundamentally solving the issue of oilstone clogging. However, there is little public information on the honing process for single-layer oilstones.

Furthermore, the valve spool reciprocates within the valve sleeve, inevitably causing wear and temperature changes. This necessitates that valve sleeve materials possess high wear resistance and low expansion coefficient [19]. Commonly used materials include die steel, martensitic stainless steel, superalloy, and bearing steel, such as Cr12MoV, 9Cr18Mo (440C), 4Cr14Ni14W2Mo, and GCr15 [20]. Currently, the most widely used material for electro-hydraulic servo valve sleeves is 9Cr18MoV. This material is modified by adding vanadium (V) elements to 9Cr18Mo, resulting in finer grains, higher wear resistance, and better toughness. The hardness of 9Cr18MoV after quenching is about HRC58. The high hardness and toughness characteristics of martensitic stainless steel will exacerbate the wear of sintered oilstone, mainly manifested as abrasive breakage, particle detachment, blockage, and thermal wear problems, adding difficulty to valve sleeve machining.

Based on an analysis of current research, it is evident that although honing processes are widely used for valve sleeves, manual lapping is still necessary to ensure final accuracy. On the other hand, some researchers have adopted the honing process as the final finishing process. However, due to the high hardness of the material, the severe wear of sintered oilstones, and the inability to ensure stable material removal, workers must frequently dress the oilstones, significantly limiting honing efficiency. Considering the single-layer tool advantages of large chip accommodation space, this study proposes to use electroplated superabrasive oilstones for the honing process of servo valve sleeves. The aim is to compare and analyze the differences in honing efficiency and surface quality between electroplated and traditional sintered oilstones, ultimately establishing an efficient and precise machining method for electro-hydraulic servo valve sleeves. Ultimately, in the fields of aerospace and automotive, the combination of electroplated oilstone and sintered oilstone optimizes processing efficiency, reduces material waste, and meets the requirements of green manufacturing.

## 2. Materials and Methods

### 2.1. Oilstone

#### 2.1.1. Oilstone Preparation

Sintered oilstone is prepared through a series of steps, including mixing, molding, and high-temperature sintering, a process that is well-established. Consequently, 230# sintered oilstones were customized.

To prepare electroplated oilstone, an electroplating device was designed, as shown in Figure 1. The preparation of electroplated oilstone requires typical composite electroplating. The effective area of nickel plate is 25 × 25 mm^2^. A power supply produced by KIKUSUI (Yokohama, Japan) is used for this process. Nickel ions are deposited onto the cathode, fixing the grains on the matrix and finally forming the electroplated oilstone.

Finally, two types of oilstone samples were prepared by electroplated and sintered processes, with the same quality of 230# CBN as the grains, as shown in Figure 2.

#### 2.1.2. Oilstone Topography

The grains of the undressed sintered oilstone were mostly wrapped in binders, as shown in Figure 3a. The bonding strength of grains depends on the thickness of the nickel binder, as shown in Figure 4b. The exposed height of electroplated oilstone grains was set at 30% of the grain size, which is significantly higher than that of sintered oilstone. Additionally, the concentration of grains obtained through the buried sand method was relatively high, and their distribution of grains was uniform, as shown in Figure 4a. To compare the concentration of electroplated and sintered oilstone grains, the number of grains per unit area was used as the evaluation index. The volume concentration of sintered oilstone was 50%, with a concentration before dressing of 18 grains/mm^2^. In contrast, the concentration of electroplated oilstone was 200 grains/mm^2^.

After dressing, the grain concentration in the electroplated oilstone showed no significant change, whereas the grain concentration in the sintered oilstone increased to 42 grains/mm^2^, which remained lower than that of the electroplated oilstone, as shown in Figure 3b.

Due to inherent defects such as low grain concentration and low exposed height, sintered oilstones are prone to clogging, leading to unstable material removal. In contrast, electroplated oilstones effectively address these issues, thereby improving the efficiency and precision of the honing process.

#### 2.1.3. Exposed Height of Oilstone Grains

To provide a more comprehensive characterization of the grain exposed height, a quantitative analysis was conducted. The method for obtaining the exposed height distribution of grains in electroplated oilstones is illustrated in Figure 5. After a series of operations, including 3D morphology data collection, surface flattening, binarization, denoising, and grain extraction, the exposed height distribution of grains was determined. Firstly, the surface morphology data matrix of the electroplated oilstone was obtained using an optical surface profilometer. To facilitate the statistics of grain height, coordinate transformation was used to flatten the curved surface of the oilstone into a plane, as shown in Figure 6a. Since the surface morphology of oilstones is essentially an image, binarization processing was employed to achieve greater contrast on the oilstone surface, as shown in Figure 6b. To eliminate noise, median filtering, Gaussian filtering, and opening operations were applied, resulting in a smoother oilstone surface. In the denoised image (Figure 6c), the white regions represent grains, and each closed loop corresponds to an individual grain. Grains were identified and numbered using the connected region function, as shown in Figure 6d, which counted the grains. By matching the processed binary images with the original data, the exposed height was determined by subtracting the average height of the surrounding area from the highest point of each region. The expression is as follows:(1)$$H_{i}=h_{i\max}-\sum_{j=1}^{n_{p}}h_{ij}/n_{p}$$
where *H_i_* is the exposed height of the *i*th grain, *h_i_*_max_ is the maximum height of the *i*th grain, *h_ij_* is the *j*th height information of the edge of the *i*th grain, and the data points of the edge of the grain are *n_p_*.

Through statistical analysis, the exposed height distribution of grains was finally obtained, as shown in Figure 7. As dressing progresses, a small number of grains with shallow buried depths fall off (Figure 8a), while some grains with higher exposed heights experience slight breakage (Figure 8b), as shown in Figure 8. The exposed height distribution of the electroplated oilstone tends to be stable, indicating that the electroplated oilstone has entered the stable wear stage and is operating in its optimal condition. By employing image processing techniques, the grain height distribution was measured with high accuracy, eliminating human statistical errors and significantly improving the analysis efficiency.

Before dressing, the grains of sintered oilstone were almost completely wrapped in binder. Although dressing exposes the grains, their exposed height remains low, generally less than 1 μm. Due to the rough surface of the sintered oilstone, existing methods cannot accurately calculate the grain’s exposed height, making quantitative analysis of the height distribution unfeasible. However, since the grains are evenly mixed during the preparation of the sintered oilstone, both the grain concentration and the exposed height distribution tend to stabilize after dressing. As a result, the grain concentration was used as an indicator to determine whether the exposed height distribution reached a stable stage and whether the dressing process was complete.

### 2.2. Plate Scratch Test

The flat scratch test was conducted on a machining center produced by DMG (Bielefeld, Germany). During the test, the spindle was locked to prevent rotation, and the oilstone fixture was connected to the spindle via the tool holder. The flat oilstone was fixed to the fixture with bolts, applying a certain pressure on the plate, and sliding along the width direction of the oilstone at a certain speed, ultimately leaving scratches on the surface of the plate, as shown in Figure 9. The plates were made of 9Cr18MoV, with their chemical composition detailed in Table 1. The coolant used was Castrol 9554, a water-based emulsion with a concentration of 5%, applied at a pressure was 4 bar. The spindle fed negatively along the z-axis, which drove the sleeve of the oilstone fixture downward, compressing the spring in the fixture. The spring’s reaction force acted on the piston, transmitting the normal force to the oilstone. The normal force applied to the oilstone was proportional to the spring’s deformation. The length and width of the flat oilstone were identical to the actual oilstone, but the trajectory interaction of the oilstone was not considered. A Kistler 9129AA dynamometer (Winterthur, Switzerland) was used to measure the normal and tangential forces during the scratch test. The groove topography of the processed surface was measured by a Sensofar optical profiler (Barcelona, Spain). The groove cross-sectional area was further processed to determine the equivalent honing depth. In the flat scratch test, the parameters to be studied included cutting speed, normal force, and overtravel. The specific parameters are listed in Table 2. For the plate scratch test, low cutting speed and sufficient cooling will not produce significant thermal effects. Moreover, the high exposure of electroplated oilstone abrasive particles will not cause a temperature rise due to blockage. Thus, the influence of cutting speed on the microstructural deformation and subsurface damage can be ignored.

The key to the plate scratch test is ensuring proper surface contact between the oilstone and plate. Therefore, both the oilstone and plate require pretreatment prior to testing. Through grinding and polishing, the surface roughness of the plate was reduced to less than 0.01 μm. The oilstone must exhibit good machining capability and flatness. Similar to actual oilstones, the flat oilstone also needs dressing. Figure 10 shows the dressing platform, where the dressing of the flat oilstone was carried out on an ultra-precision grinder (Xi’an, China) using a 1500-grit resin-based CBN wheel.

To quantitatively evaluate the actual honing depth, the equivalent honing depth was used as the evaluation index, which refers to an imaginary cross-sectional thickness gathered by each chip section cut by grains within the contact length per unit width of oilstone [21]. That is, the ratio of the total cutting cross-sectional area to the length of the oilstone engaged in honing. First, an optical profiler was used to capture all the scratch information on the plate, as shown in Figure 11. Under constant processing parameters, the cross-sectional profile in the direction of the oilstone’s movement remained consistent. Therefore, the material removal volume (MRV) from the starting point to the end point of the flat oilstone can be expressed as the sum of the MRV of all the scratches, which is calculated as follows:(2)$$V_{\mathrm{total}}=\sum_{i-1}^{m}V_{i}=\sum_{i-1}^{m}A_{i}l_{m}=\sum_{i-1}^{m}(A_{\mathrm{groove}i}-A_{\mathrm{uplift}i})l_{m}=A_{\mathrm{total}}l_{m}$$
where *m* represents the number of scratches, *l*_m_ represents the moving distance of the oilstone, *A*_groove_ is the cross-sectional area of the grooves, *A*_uplift_ is the cross-sectional area of the uplifts, and *A*_total_ is the total cross-sectional area of removed material.

The equivalent honing depth can be expressed as the ratio of the total cross-sectional area of removed material to the actual contact length of the oilstone, which is
(3)$$a_{e}=\frac{V_{\mathrm{total}}}{l_{m}l_{c}}=\frac{\sum_{i=1}^{m}\left(A_{\mathrm{groove}i}-A_{\mathrm{uplif}ti}\right)}{l_{c}}=\frac{A_{\mathrm{total}}}{l_{\mathrm{stone}}-l_{\mathrm{ot}}}$$
where *l*_c_ is the actual contact length between the oilstone and the plate, *l*_stone_ is the length of the oilstone, and *l*_ot_ is the overtravel.

The total cross-sectional area indirectly reflects the material removal ability. Furthermore, the cross-sectional area in honing plays a crucial role in determining the frictional properties of the machined surface. A larger groove area can usually increase the retention of lubricant, reduce friction, and enhance wear resistance and heat dissipation. However, excessively large grooves can lead to fluid dynamics instability and surface roughness.

### 2.3. Honing Test

The honing test was conducted on a DMG machining center, featuring a maximum spindle speed of 42,000 rpm, a maximum reciprocating speed of 40 m/min, runout within 5 μm, and a maximum internal cooling system pressure of 4 MPa. A custom-made single oilstone honing tool, designed for pressure honing, was used. Coolant was delivered through the spindle’s internal cooling system, enabling radial feeding of the oilstone. Two types of oilstones were used: a 230# sintered oilstone and a 230# electroplated oilstone. The workpiece material was 9Cr18MoV, with a quenched hardness of HRC58. The diameter of the pre-honed workpiece ranged from 6.965 to 6.975 mm, and the hole length was 60 mm. The experimental setup is shown in Figure 12.

The hole diameters before and after processing were measured in situ using pneumatic gauge, with an accuracy of 0.3 μm. The probe was fixed to the spindle via a tool holder. Different depths of diameter were achieved by moving the spindle in the z-direction.

To ensure good cylindricity of the honed hole, the setting of the upper and lower overtravel is crucial, as shown in Figure 13, where *h*_valve_ is the hole length, *d*_0_ is the pre-honed hole diameter, *l*_stone_ is the oilstone length, *l*_up_ is the upper overtravel, and *l*_down_ is the lower overtravel.

When the honing pressure *P* remains constant, the normal force *F_n_* acting on the surface of the valve sleeve also remains constant. According to Equation (2), the larger the overtravel, the greater the equivalent honing depth, which causes the honing depth at both ends of the valve sleeve to be greater than that in the middle. Furthermore, under high honing pressure, this phenomenon is more pronounced. Additionally, the contact time between the oilstone and the workpiece varies at different hole depths. Therefore, using the ratio of the total MRV to time to represent the relationship between material removal rate (MRR) and honing pressure is inaccurate. The honing MRR is defined as the MRV per unit time [22,23], or the MRV per unit time and per unit contact area [24], which does not account for the constantly changing contact state between the oilstone and the workpiece, thus only reflecting the overall material removal capability of the oilstone, rather than accurately representing the material removal situation at different hole depths.

Based on the above analysis, to establish the relationship between MRR and honing pressure, it is necessary to eliminate the interference caused by varying overtravel and contact times. When the actual hole length is greater than twice the oilstone length, as shown in Figure 14, the entire oilstone remains in contact with the inner wall of the hole between points *z*_1_ and *z*_2_, thus avoiding the impact of overtravel and contact times. Therefore, a method for evaluating the material removal capability of honing was established, defined as the MRR per unit width, providing a basis for the quantitative calculation of subsequent MRV distribution along the hole depth direction. Thus, the honing MRR per unit width within the range from *z*_1_ to *z*_2_ can be expressed as
(4)$$Q_{w}=\frac{A_{w}}{2n_{h}t}$$
where *A_w_* is the MRV per unit width, *n*_h_ is the strokes of the honing tool, and *t* is the honing time for one stroke at the hole depth *z*. A stroke refers to the movement of the oilstone from the upper limit to the lower limit.

The inner hole of the valve sleeve is cylindrical, and the cross-section at any depth is circular, as shown in Figure 7. The MRV per unit width can be expressed as
(5)$$A_{w}=\frac{1}{4}\pi (d_{1}{\left(z\right)}^{2}-d_{0}{\left(z\right)}^{2})$$
where *d*_1_(*z*) is the honed bore diameter at depth *z* and *d*_0_(*z*) is the initial bore diameter at depth *z*. For the honing tool used in the test, the length of oilstone was 24 mm. As shown in Figure 15, the entrance end face of the hole was taken as the xy-plane, and the z-axis was oriented downward perpendicular to the entrance end face. According to Figure 14, considering oilstone length and hole length, in the depth range from 24 to 36 mm, the material removal is not affected by the overtravel and contact time.

According to Figure 6, the honing time t for one stroke at the hole depth *z* can be expressed as
(6)$$t=\frac{l_{\mathrm{stone}}}{v_{a}}$$

Combining Equation (3) with Equation (5), the MRR per unit width can finally be expressed as
(7)$$Q_{w}=\frac{\pi v_{a}(d_{1}{\left(z\right)}^{2}-d_{0}{\left(z\right)}^{2})}{8n_{h}l_{\mathrm{stone}}}$$

## 3. Results and Discussion

### 3.1. Equivalent Honing Depth

#### 3.1.1. Honing Speed in Plate Scratch Test

In the traditional metal cutting theory, cutting speed affects the cutting heat and strain rate, which in turn affects the cutting force and the plastic flow of the material in the cutting area. When the honing width equals the length of the oilstone, the equivalent honing depth at different honing speeds can be calculated using Equation (3), as shown in Figure 16. For electroplated oilstone subjected to 40 N normal force, as the honing speed increases from 1 to 30 m/min, the equivalent honing depth fluctuates around 0.012 μm. In contrast, for the sintered oilstone under a normal force of 120 N, the honing speed also increases from 1 to 30 m/min, but the equivalent honing depth consistently fluctuates around 0.00031 μm.

Force ratio, defined as the ratio of normal force to tangential force during grinding, is a physical quantity to evaluate the difficulty of removing the material. A larger force ratio indicates greater difficulty in material removal. For the same oilstone, under the same normal force, honing speed basically has no influence on the force ratio, so the force ratio is basically kept unchanged. The force ratio of 230# electroplated oilstone is 2.9, while that for the 230# sintered oilstone is 6.15, as shown in Figure 17. This suggests that honing speed does not significantly affect the equivalent honing depth, and it is easier to remove material with the electroplated oilstone.

#### 3.1.2. Normal Force

The radial feed of constant pressure honing is closely related to the normal force acting on the oilstone. With a honing speed of 5 m/min and zero overtravel, the equivalent honing depth increases as the normal force rises. For the electroplated oilstone, when the normal force increases from 20 to 200 N, the equivalent honing depth increases from 0.0067 to 0.079 μm. In comparison, for the sintered oilstone, as the normal force increases from 20 to 480 N, the equivalent honing depth increases from 0.000047 to 0.0017 μm, (Figure 18). Whether electroplated or sintered oilstone, the equivalent honing depth is basically linear with the normal force, with the slope for electroplated oilstone being significantly steeper than that of sintered oilstone. This is attributed to the higher grain exposed height and concentration of the electroplated oilstone, which results in a larger equivalent honing depth and greater material removal. In addition, the tangential force increases with normal force, but for the same oilstone, the force ratio basically remains unchanged, as shown in Figure 19. Within the normal force variation range of honing, the contact state between the oilstone and the workpiece remains unchanged, leading to a stable force ratio.

#### 3.1.3. Overtravel

Overtravel is a key parameter in honing that directly affects the shape accuracy of the valves. The honing speed is 5 m/min and normal force is 80 N. For constant pressure honing, the MRV at the inlet and outlet of the valve sleeve is affected by the contact time and contact pressure, with the contact pressure being directly impacted by the overtravel. Equation (3) shows that the factors affecting the equivalent cutting depth include the number of scratches, the overtravel, and the cross-sectional area of the material removed by the scratches. When pressure is constant, the number of scratches is slightly reduced as the overtravel increases, while the depth of a single scratch slightly increases. However, the final total cross-sectional area of removed material basically remains essentially unchanged (Figure 20). Under the same normal force, the overall scratch depth of the grain increases with increasing of the contact pressure.

The functional image of the equivalent honing depth with the overtravel can be obtained by Equation (3), as shown in Figure 20. From the functional expression, the equivalent honing depth is inversely proportional to the overtravel. Furthermore, the inverse proportional relationship can be approximated as a linear relationship due to the small variation range of the overtravel. For the electroplated oilstone, the total cross-sectional area of removed material fluctuated slightly around 699.12 μm^2^, while for sintered oilstone, it fluctuated around 6.08 μm^2^. When the overtravel increases from 0 to 8 mm, the equivalent honing depth increases from 0.029 to 0.039 μm for electroplated oilstone and from 0.00025 to 0.00036 μm for sintered oilstone (Figure 21). Under the optimal condition, the MRR of electroplated oilstone is about 2.5 times that of sintered oilstone.

### 3.2. Material Removal Rate

The direct machining parameters of constant pressure honing mainly include honing pressure, spindle speed, and reciprocating speed. Klocke found that the cross-hatch angle has a great influence on the honing material removal [25]. Both spindle speed and reciprocating speed have an impact on the cross-hatch angle, as described in Equation (8). In addition, the honing speed is described in Equation (9), which is another indirect processing parameter of honing that affects the honing time. Therefore, the cross-hatch angle and honing speed were used as processing parameters. To explore the influence of main factors on the MRR of valve sleeve, three groups of single-factor tests were carried out with 60 strokes. Other parameters are summarized in Table 3.
(8)$$\theta =2\arctan(\frac{v_{s}}{v_{a}})=2\arctan(\frac{\pi d_{\mathrm{tool}}n_{s}}{1000v_{a}})$$
(9)$$v=\sqrt{{v_{s}}^{2}+{v_{a}}^{2}}=\sqrt{{\left(\frac{\pi d_{\mathrm{tool}}n_{s}}{1000}\right)}^{2}+{v_{a}}^{2}}$$
where *θ* is the cross-hatch angle, *v* is the honing speed, *v_s_* is the spindle speed, *v_a_* is the reciprocating speed, *n_s_* is the spindle speed, and *d*_tool_ is the diameter of the honing tool.

#### 3.2.1. Honing Pressure

Honing pressure is one of the key factors affecting the MRR. It directly influences the normal force acting on the workpiece. As analyzed in Section 3.1, the normal force directly affects the equivalent honing depth, which ultimately impacts the MRV.

Only by overcoming the piston friction and spring reaction force of the hydraulic cylinder inside the honing tool can the push cone push the oilstone to move radially, as shown in Figure 4. Consequently, the oilstone will only expand radially and contact the workpiece to achieve material removal when the honing pressure exceeds a certain critical value. As shown in Figure 22, the axial honing force is linearly positively correlated with the honing pressure, with the function graph intersecting the y-axis at approximately 1 bar, termed the critical honing pressure. Therefore, the axial honing force is greater than zero only when the honing pressure exceeds 1 bar, indicating that the oilstone starts to contact the workpiece. In summary, the selected honing pressure should be at least greater than 1 bar. Theoretically, as long as the oilstone thickness and the hole diameter to be machined are consistent, the critical honing pressure should remain the same, whether using electroplated or sintered oilstone.

The diameter change of the honed hole indirectly reflects the MRV. Honing was performed using a 230# sintered oilstone at a cross-hatch angle of 22.6° and a honing speed of 25.5 m/min. As shown in Figure 22, the effect of honing pressure on the change in hole diameter reveals that when the honing pressure increases from 2 bar to 4.5 bar, the change in hole diameter initially increases, then decreases. The change in hole diameter increases from 0.0011 mm at 2 bar to 0.0049 mm at 4 bar, due to the increase in normal force caused by the higher honing pressure, which leads to a greater equivalent honing depth, ultimately resulting in a larger MRV. However, when the honing pressure exceeds 4 bar, the MRV begins to decrease. The exposed height of the sintered oilstone grains is relatively low. As the honing pressure increases, the equivalent honing depth also increases, reducing the chip accommodation space. Consequently, the generated chips cannot be expelled promptly and tend to adhere to the oilstone surface. With the increasing number of strokes, chip adhesion becomes more severe, diminishing the oilstone grinding capability, thus leading to a decreasing trend in the change of hole diameter. According to Equation (7), the influence of honing pressure on the MRR per unit width can be derived, as shown in Figure 23, which exhibits the same trend as the hole diameter change. As the honing pressure increases from 2 to 4 bar, the MRR per unit width increases from 0.015 to 0.093 mm^3^·min⁻^1^·mm⁻^1^. However, when the honing pressure further increases from 4 to 4.5 bar, the MRR per unit width starts to decrease from 0.093 to 0.078 mm^3^·min⁻^1^·mm⁻^1^, which is consistent with the trend and reason for the diameter change. There are two methods to increase the upper limit of honing pressure: improving the oilstone structure to increase more chip pace and improving the cooling and lubrication structure to remove chips.

For electroplated oilstones, as shown in Figure 24, the effect of honing pressure on the hole diameter change shows that, under the same cross-hatch angle and honing speed as the sintered oilstone, the honing pressure increases from 1.8 to 2.8 bar, and the hole diameter change increases from 0.0032 to 0.018 mm. As shown in Figure 19, the force ratio between a 230# sintered oilstone is more than twice that of a 230# electroplated oilstone, which can lead to increased oilstone wear, affecting its lifespan and processing stability. According to Equation (7), the effect of honing pressure on the MRR per unit width can be obtained shown in Figure 24, showing the same trend as the hole diameter change. As the honing pressure increases from 1.8 to 2.8 bar, the MRR per unit width increases from 0.048 to 0.32 mm^3^·min⁻^1^·mm⁻^1^. The MRR per unit width is significantly higher for electroplated oilstone compared to sintered oilstone.

#### 3.2.2. Cross-Hatch Angle

Previous studies have shown that the honing cross-hatch angle affects honing force, oilstone self-sharpening, and contact conditions, thereby significantly impacting honing efficiency.

For sintered oilstones with a honing speed of 25 m/min, as shown in Figure 25, the influence of the cross-hatch angle on diameter change shows that increasing the cross-hatch angle from 16 to 36° results in a decreasing trend in diameter change. Initially, as the cross-hatch angle increases from 16 to 32°, there is a roughly linear downward trend. When the honing pressure is 3 bar, the diameter change decreases by 0.35 μm for every 4° increase. However, when the cross-hatch angle exceeds 32°, the decreasing trend in diameter change accelerates, resulting in a reduction of 1.01 μm for every 4° increase. As the cross-hatch angle increases, the honing trace becomes sparse. As shown in Figure 26, honing trace at a cross-hatch angle of 22.6° (Figure 26(left)) is sparse at a cross-hatch angle of 39.6° (Figure 26(right)). The contact trace between the oilstone and the workpiece becomes shorter, resulting in a reduction in the MRV and eventually a gradual decline in the diameter change.

As the cross-hatch angle increases from 16 to 36°, the MRR per unit width initially increases, then decreases, as shown in Figure 25. The material removal is stable, with the cross-hatch angle at the inflection point of MRR per unit width occurring around 32°. This inflection point remains consistent across different honing pressures. An increase in the cross-hatch angle results in a decrease in the MRV, but an increase in the cross-hatch angle results in an increase in the reciprocating speed, resulting in a reduction in honing time, as shown in Figure 26. For a honing pressure of 3 bar, when the cross-hatch angle increases from 16 to 32°, the MRR per unit width increases from 0.031 to 0.054 mm^3^·min^−1^·mm^−1^. The reduction of honing time dominates and finally the MRR per unit width increases. When the cross-hatch angle further increases from 32 to 36°, the MRR per unit width decreases from 0.054 to 0.034 mm^3^·min^−1^·mm^−1^. In this case, the decrease in MRR is more significant, outweighing the reduction in honing time, which finally leads to the decrease of MRR per unit width. Thus, to ensure a high MRR, the cross-hatch angle is about 32°.

For electroplated oilstone (Figure 27), the influence of cross-hatch angle on the MRR per unit width follows a similar trend to that of sintered oilstones, with the inflection point also occurring at 32°.

#### 3.2.3. Honing Speed in Honing Test

Honing speed is the combined spindle speed and reciprocating speed. Under a certain number of strokes, a faster honing speed results in a shorter honing time. Therefore, honing speed is a key factor affecting machining efficiency.

The honing speed is increased from 10 to 35 m/min by adopting 230# sintered oilstone and keeping a cross-hatch angle of 24° and a honing pressure of 3 bar, and the diameter change basically keeps unchanged, which is consistent with the results of plate scratch test in Section 3.1.

On one hand, due to the characteristics of the honing process, the oilstone is embedded in the groove, with a certain gap. Compared to the sintered oilstone at a honing speed of 25 m/min, excessive honing speed (*v* = 30 m/min) can cause the oilstone to experience impacts, resulting in breakage at both ends of the oilstone, as shown in Figure 28, affecting the honing stability, making the error bar of the MRR per unit width bigger and intensifying material removal instability, as shown in Figure 29. The honing axial force at the two honing speeds reveals that the axial force at 25 m/min shows a consistent trend across strokes. However, at the honing speed of 30 m/min, the variation trend of axial force in strokes is inconsistent, as depicted in Figure 28. The breakage occurring at both ends of the oilstone alters the contact between the oilstone and the workpiece, leading to fluctuations in axial force that ultimately affect the stability of material removal. Based on the analysis of oilstone state and the characteristics of spindle acceleration and deceleration, excessive honing speed will lead to unstable material removal. For sintered oilstone, it is advisable to set the honing speed at 25 m/min to ensure both high processing efficiency and honing stability.

For electroplated oilstone, the substrate is die steel, which has higher strength and impact resistance than sintered oilstone. The electroplating process ensures a strong bond between the grains and the substrate, making the electroplated oilstone less prone to breaking during honing, unlike sintered oilstone, as shown in Figure 30. Comparing Figure 30a,b, there is no significant difference in the morphology of electroplated oilstone before and after honing. Therefore, electroplated oilstones can operate at higher honing speed than sintered oilstones. When the honing speed is 30 m/min, the stability of diameter change can be maintained, as shown in Figure 31. For the machining center, the acceleration of spindle in the z-direction is fixed. When the reciprocating speed increases from 3 to 7 m/min, the time required to reach the set speed increases from 87 to 110 ms. Higher honing speeds result in a greater z-direction displacement upon reaching the set speed (as illustrated in Figure 32). The cross-hatch angle is constantly changing, resulting in unstable material removal at the hole inlet and outlet. When the honing speed exceeds 30 m/min, the unstable length exceeds 3.5 mm, exceeding 1/10 of the hole length, negatively affecting shape accuracy and surface morphology consistency. Therefore, by weighing precision and efficiency, the optimal honing speed for electroplated oilstone is determined to be 30 m/min.

### 3.3. Surface Quality

According to Section 3.2, the optimal honing speed and cross-hatch angle have been determined, so the processing parameter only consider the honing pressure. The optimal honing pressure is from 2 to 3.5 bar for 230# sintered oilstone, and the optimal honing pressure is from 1.8 to 2.55 bar for 230# electroplated oilstone. This section explores the influence of honing pressure and oilstone type on the surface morphology and roughness of electro-hydraulic servo valve sleeve. The test parameters are shown in Table 4.

As shown in Figure 33, the surface morphology of the valve sleeve after honing with 230# sintered oilstone and electroplated oilstone under different honing pressures is presented. The exposed grains on the surface of the sintered oilstone are minimal. When the honing pressure is less than 3 bar, there is no obvious difference in the workpiece topography. When the honing pressure is 3.5 bar, there is a violent extrusion between the oilstone and the workpiece, resulting in severe wear of the bond and grain detachment, which leaves deeper scratches on the valve and deteriorates the surface quality. However, compared with sintered oilstone, electroplated oilstone with the same mesh has higher exposed height and grain concentration, so the honed surface of valve sleeve with electroplated oilstone has more intensive scratches, and the depth of scratches is greater than that of sintered oilstone. To summarize, small honing pressure and sintered oilstone are conducive to obtaining high surface quality.

The influence of honing pressure and oilstone type on surface roughness is shown in Figure 34. With the increase of honing pressure, the surface roughness of both electroplated oilstone and sintered oilstone increases. For 230# sintered oilstone, when the honing pressure increases from 2 to 3 bar, the surface roughness remains stable at about 0.066 μm. However, at a honing pressure of 3.5 bar, the surface roughness increases to 0.147 μm. In the case of the 230# electroplated oilstone, as the honing pressure increases from 1.8 to 2.55 bar, the surface roughness escalates from 0.104 to 0.213 μm. According to the processing requirements of the electro-hydraulic servo valve sleeve, the desired surface roughness should be less than 0.1 μm. Although higher processing efficiency can be obtained by honing with 230# electroplated oilstone, the surface roughness of the valve sleeve honed with 230# electroplated oilstone cannot meet the required standards. Additionally, the honing pressure for the 230# sintered oilstone should not exceed 3 bar. To achieve a balance between surface quality and efficiency, it is advisable to allocate the processing allowance strategically. First, electroplated oilstone should be used for rough honing to remove large amount of material. Then, there should be a switch to sintered oilstone for fine honing to ensure the desired surface roughness.

### 3.4. High Efficiency Honing of Valve Sleeve

The valve sleeve utilizes a floating fixture, as shown in Figure 12. The hole of electro-hydraulic servo valve sleeve is 52.6 mm in length and 6.986 mm in diameter, with a cylindricity error of 11.6 μm. The target diameter is 7 mm.

The difference between the target diameter and the initial diameter is about 0.014 mm. If 230# sintered oilstone is directly used for honing, although the roughness can meet the requirements, the efficiency is low. Conversely, while using the 230# electroplated oilstone enhances efficiency, the roughness fails to meet the required standards. Therefore, to balance quality and efficiency, and improve efficiency as much as possible on the premise of meeting quality, the processing of the electro-hydraulic servo valve sleeve is divided into two steps. The specific allowance distribution and processing parameters are outlined in Table 5. According to previous research, in order to ensure the stability and efficiency of material removal, the cross-hatch angle is 32°. To ensure shape accuracy, the overtravel is 8 mm.

The diameter of valve sleeve is shown in Figure 35. The measured cylindricity error is 0.0041 mm, and the surface roughness is 0.066 μm. The processing time of sintered oilstone for the same allowance is 264 s, while the processing time outlined in Table 6 is shortened to 101 s. This indicates that the processing efficiency has increased by 1.6 times, confirming the advantages of using electroplated oilstone.

This research addressed the issues of low efficiency and instability in traditional honing by developing electroplated oilstones and their corresponding honing process. It has been found that the electroplated oilstone offers higher processing efficiency; however, the surface quality is inferior compared to the sintered oilstone of the same mesh. By combining both types of oilstones, it is possible to ensure processing quality while improving honing efficiency.

## 4. Conclusions

To explore the potential of electroplated oilstone in honing 9Cr18MoV, this study discusses and analyzes the oilstone topography, equivalent honing depth, material removal rate, and surface quality. Finally, the honing process of valve sleeve was verified, confirming the advantages of electroplated oilstone. The specific conclusions are as follows:A characterization method of oilstone honing depth was established, clarifying the influence of honing parameters on equivalent honing depth using a plate scratch test. A model relating equivalent honing depth to normal force and overtravel was developed.An evaluation method of honing material removal capability was established, defined as the MRR per unit width. For 230# sintered oilstone, the optimal honing pressure ranges from 2 to 3.5 bar, the optimal honing cross-hatch angle is 32°, and the optimal honing speed is 25 m/min. For 230# electroplated oilstone, the optimal honing pressure ranges from 1.8 to 2.55 bar, the optimal honing cross-hatch angle is 32°, and the optimal honing speed is 30 m/min. Under the optimal conditions, the MRR of electroplated oilstone is about 2.5 times that of sintered oilstone.High honing pressure combined with electroplated oilstone is conducive to obtaining large material removal rate, while low honing pressure combined with sintered oilstone is favorable for obtaining high surface quality. As honing pressure increases, surface roughness also increases. For 230# sintered oilstone, the honing pressure increases from 2 to 3 bar, and the surface roughness Ra is maintained at about 0.066 μm. However, at 3.5 bar, the Ra increases to 0.147 μm, not meeting requirement for 0.1 μm. For 230# electroplated oilstone, as honing pressure increases from 1.8 to 2.55 bar, Ra increases from 0.104 to 0.213 μm, exceeding the 0.1 μm requirement. Although honing with the 230# electroplated oilstone results in higher processing efficiency, the surface roughness of the valve sleeve does not meet the processing standards.To balance surface quality and processing efficiency for the servo valve sleeve, and reasonably allocate the processing allowance, it is recommended to first use the electroplated oilstone to remove a large amount of material, followed by the sintered oilstone to ensure surface quality. This approach increased the processing efficiency of the valve sleeve sample by 1.6 times, confirming the advantages of electroplated oilstone.Despite the findings, limitations remain in the study of electroplated oilstones. To enhance the honing surface quality of electroplated oilstone, further research will focus on the dressing method of electroplated oilstone and the preparation of finer oilstones. To explore the applicability of electroplated oilstone, studies on oilstone wear will be conducted for various materials, including cast iron, stainless steel, nonferrous metals, and hard brittle materials. Furthermore, if the honing pressure can be adjusted in real time, we can control the material removal in the depth direction to obtain the desired shape, which will greatly broaden the application scenarios of honing, such high precision irregular hole.

## Figures and Tables

**Figure 1 materials-17-06170-f001:**
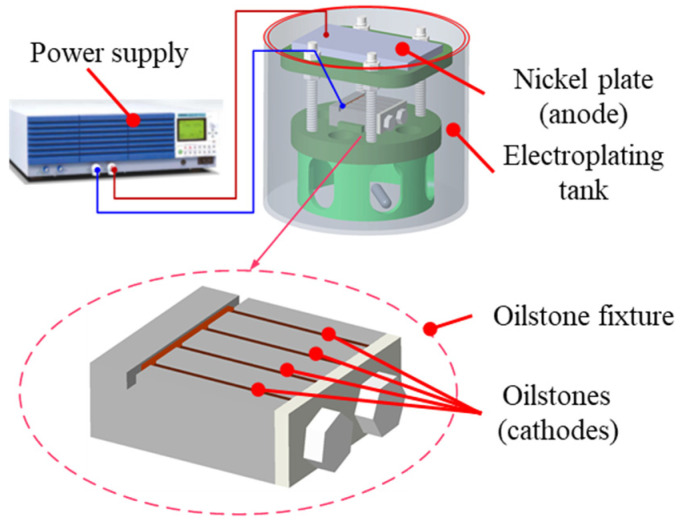
Oilstone electroplating device.

**Figure 2 materials-17-06170-f002:**
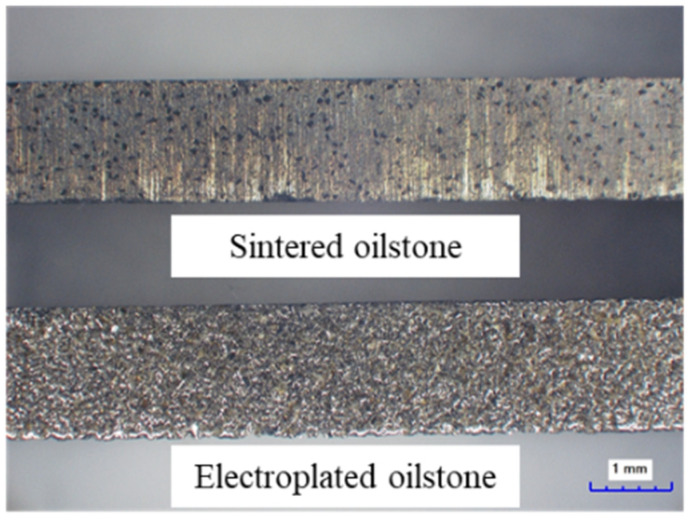
Sintered and electroplated oilstone samples.

**Figure 3 materials-17-06170-f003:**
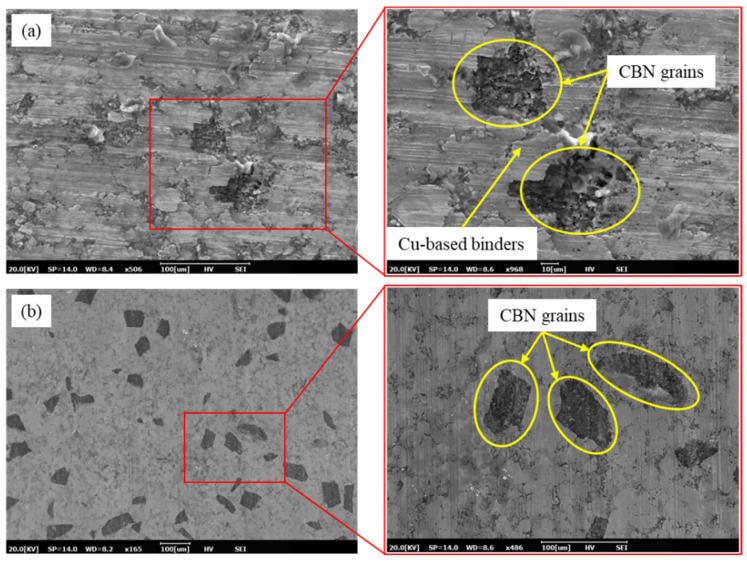
Sintered oilstone topography ((**a**) before dressing, (**b**) after dressing).

**Figure 4 materials-17-06170-f004:**
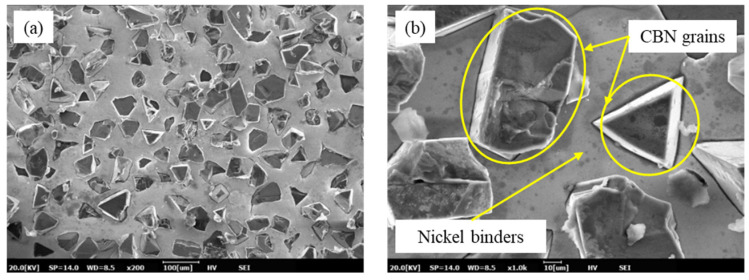
Electroplated oilstone topography ((**a**) SEM image at 200×, (**b**) SEM image at 1000×).

**Figure 5 materials-17-06170-f005:**
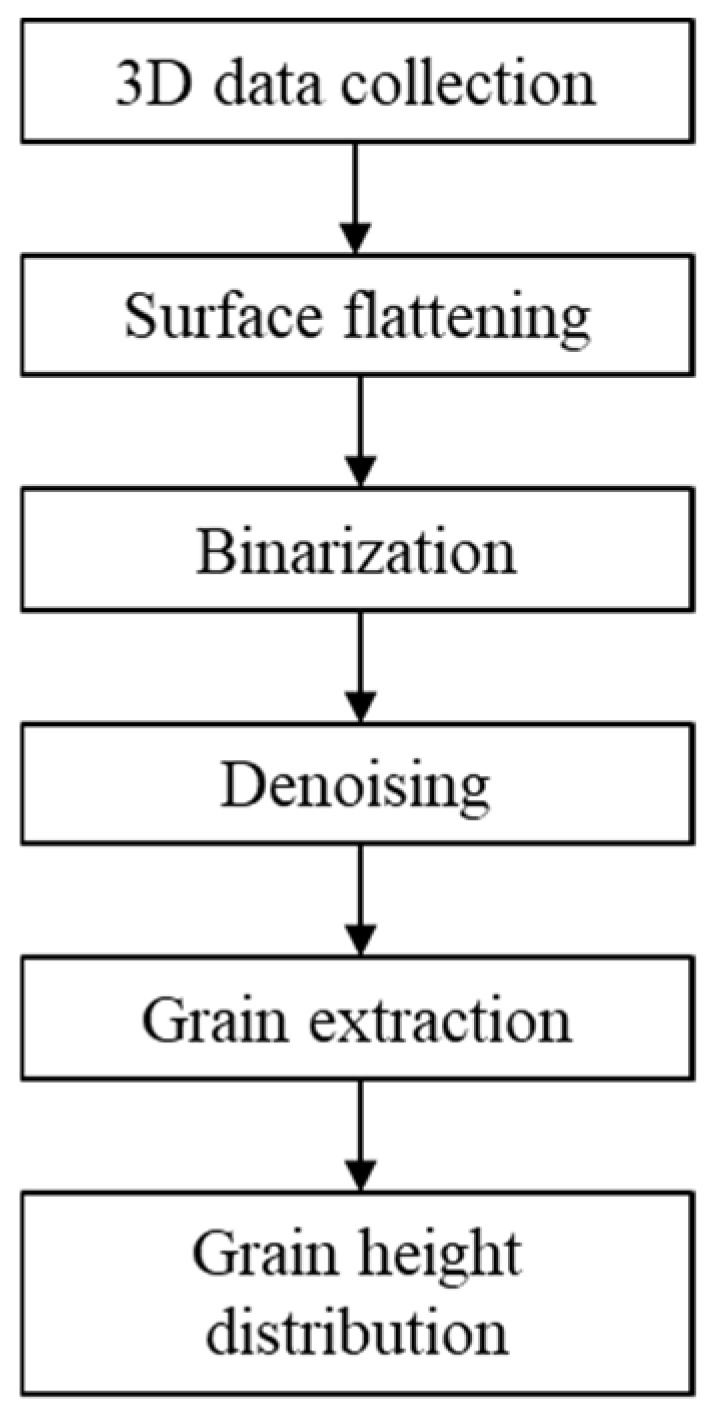
Flow chart for acquiring exposed height of grains.

**Figure 6 materials-17-06170-f006:**
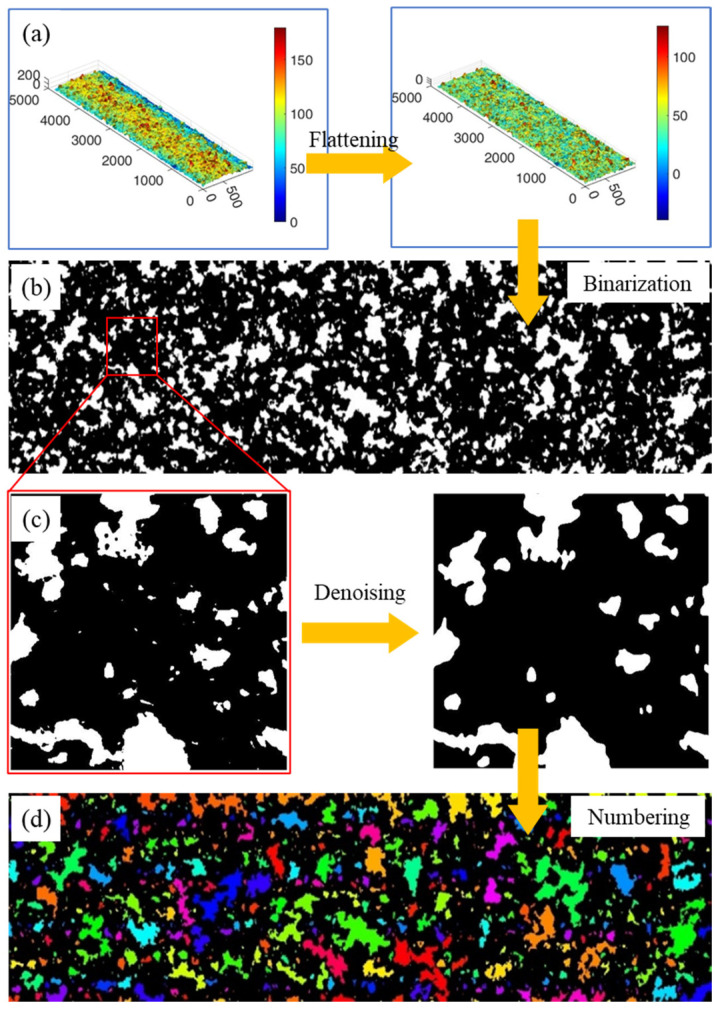
Diagram for obtaining exposed height of electroplated oilstone grains ((**a**) Falattening, (**b**) Binarization, (**c**) Denoising, (**d**) Numbering).

**Figure 7 materials-17-06170-f007:**
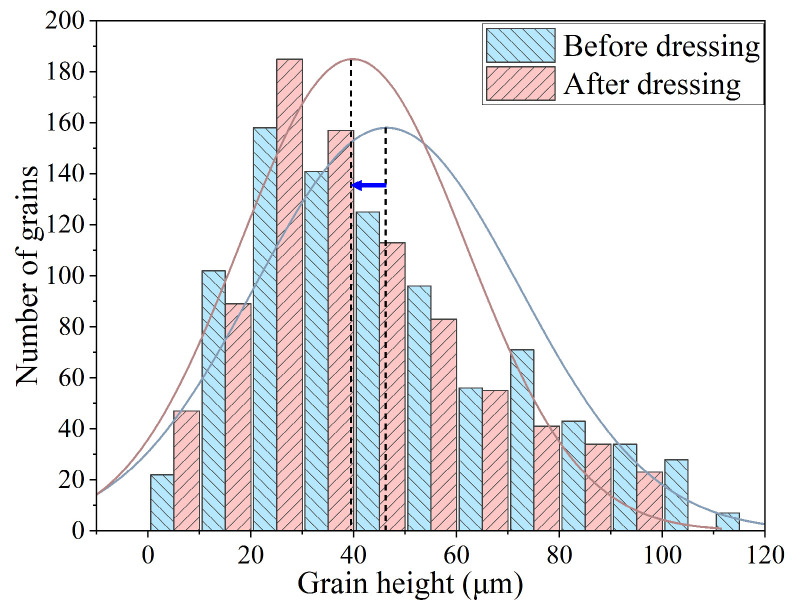
Grain height distribution of electroplated oilstone before and after dressing.

**Figure 8 materials-17-06170-f008:**
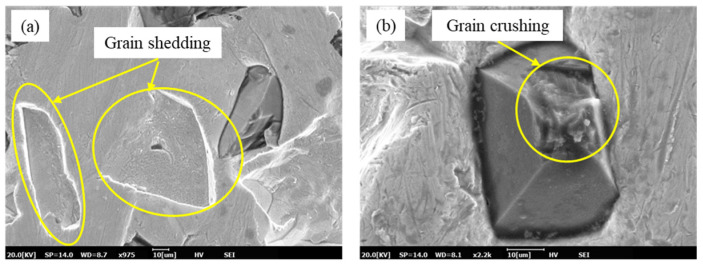
Shedding and crushing of grains ((**a**) Grain shedding, (**b**) Grain crushing).

**Figure 9 materials-17-06170-f009:**
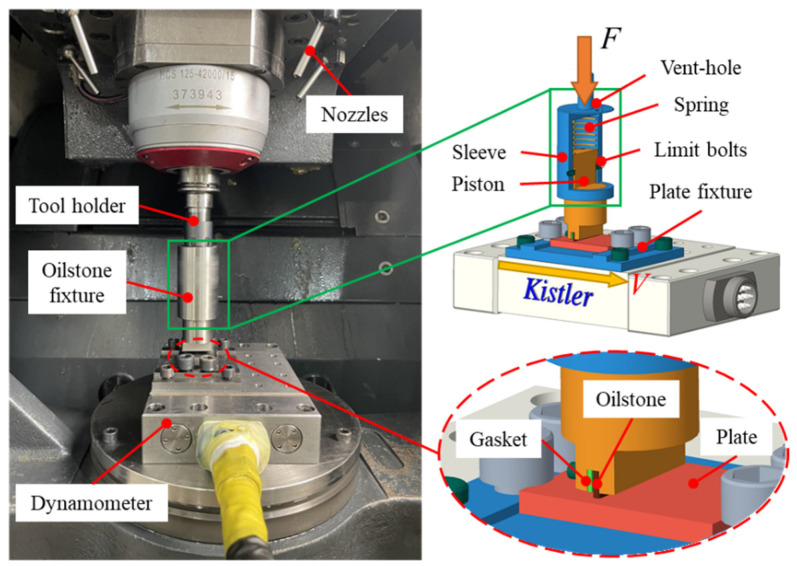
Plate scratch test platform.

**Figure 10 materials-17-06170-f010:**
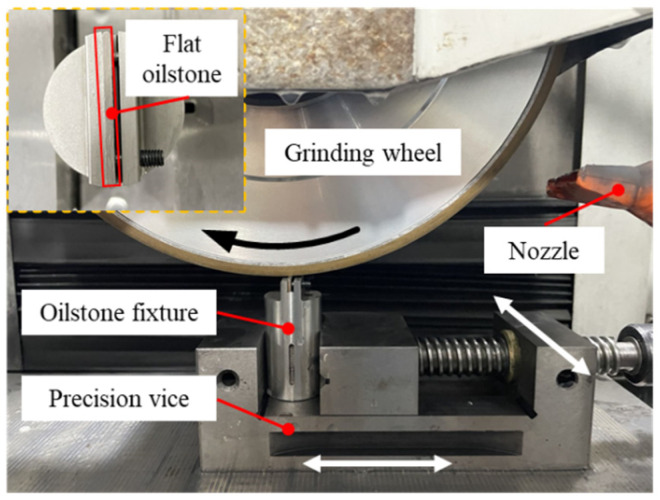
Oilstone dressing platform.

**Figure 11 materials-17-06170-f011:**
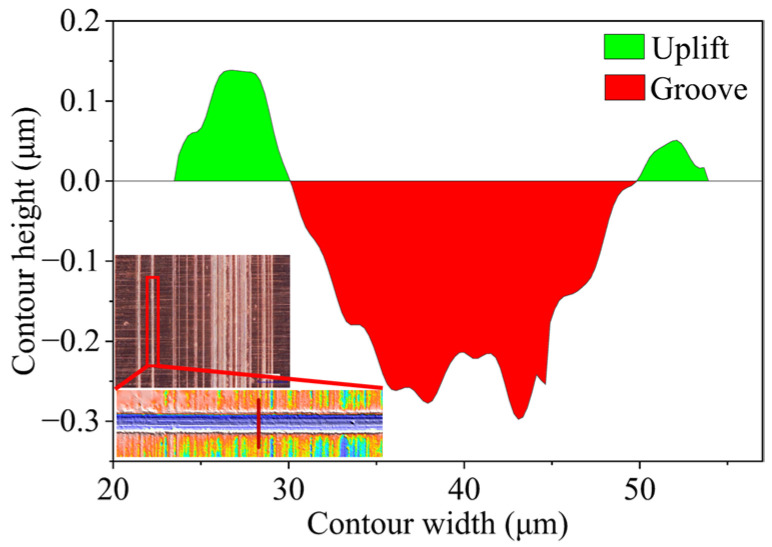
Cross-sectional profile of the scratches.

**Figure 12 materials-17-06170-f012:**
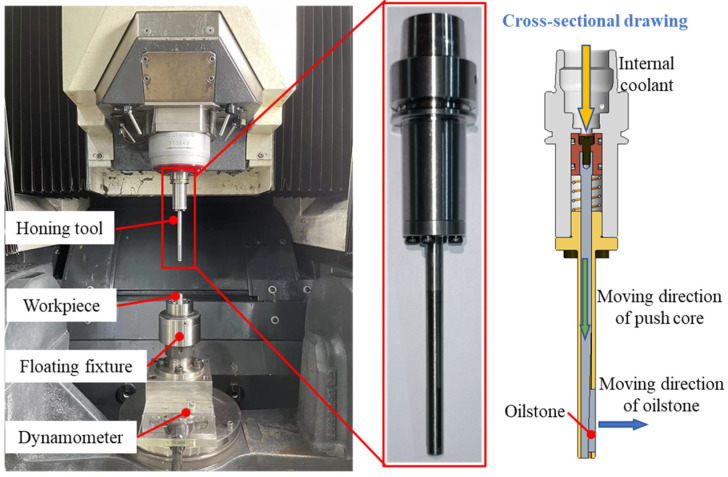
Honing platform.

**Figure 13 materials-17-06170-f013:**
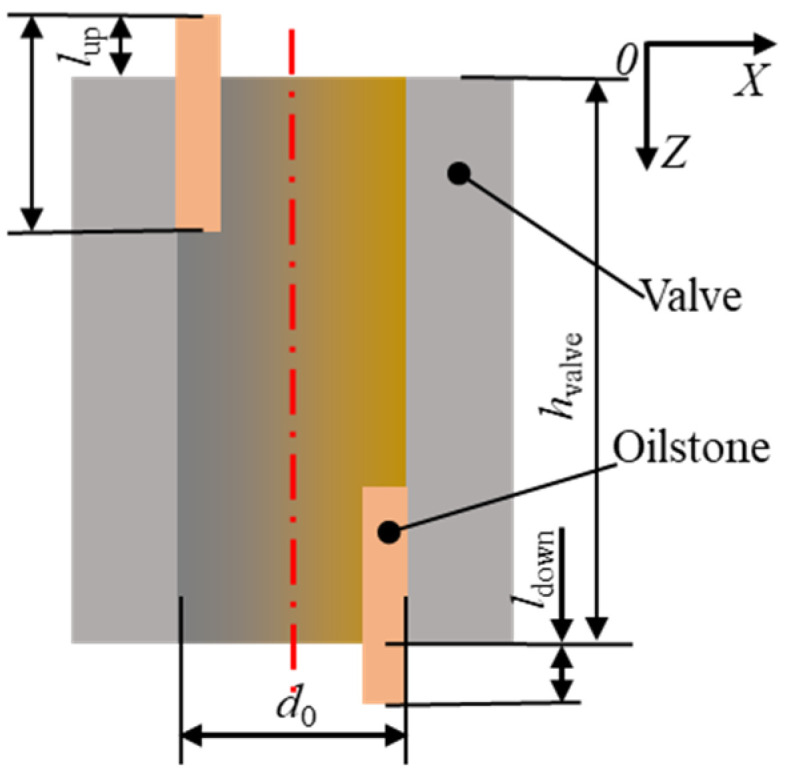
Diagram of overtravel.

**Figure 14 materials-17-06170-f014:**
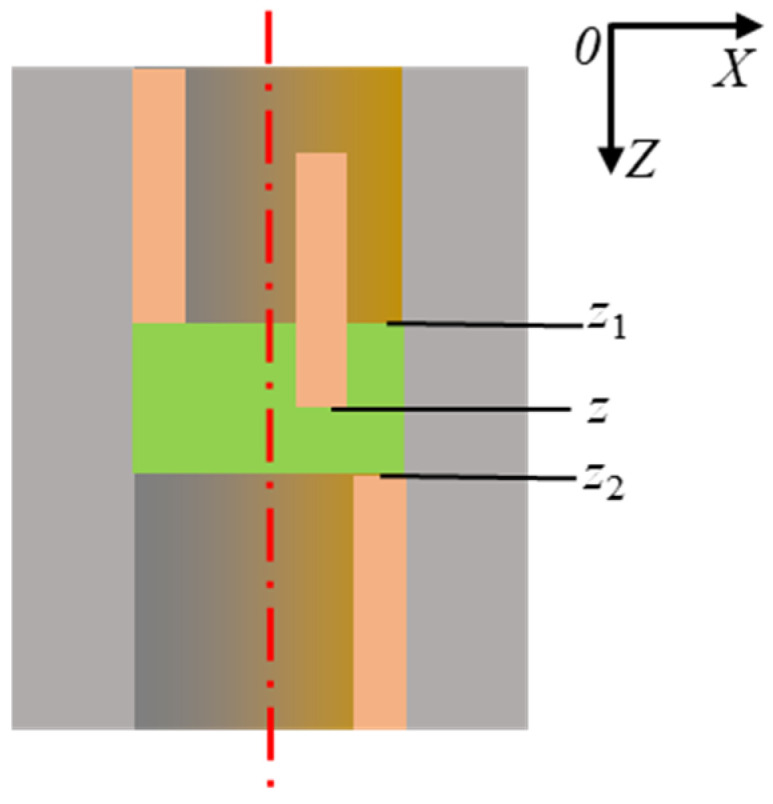
Calculation area for MRR.

**Figure 15 materials-17-06170-f015:**
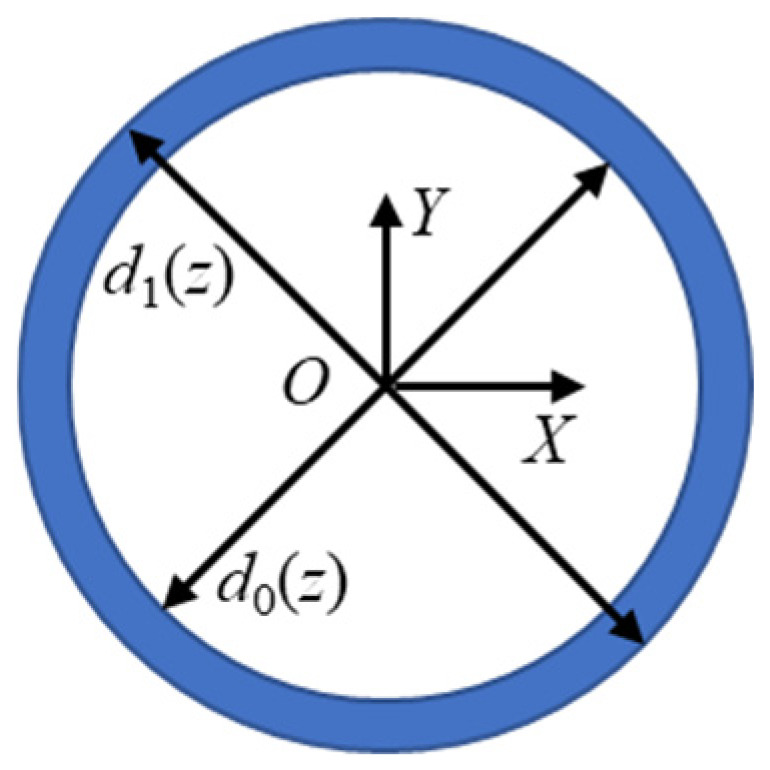
Diagram of MRV per unit width.

**Figure 16 materials-17-06170-f016:**
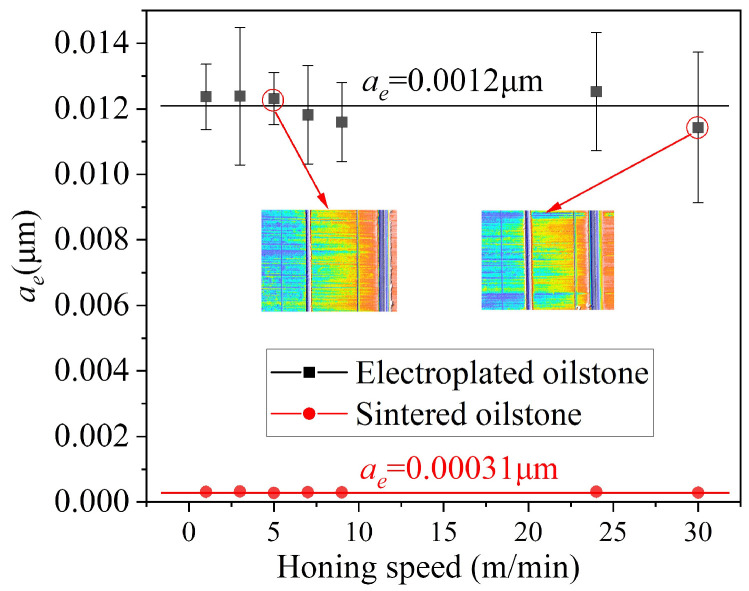
Influence of honing speed on equivalent honing depth.

**Figure 17 materials-17-06170-f017:**
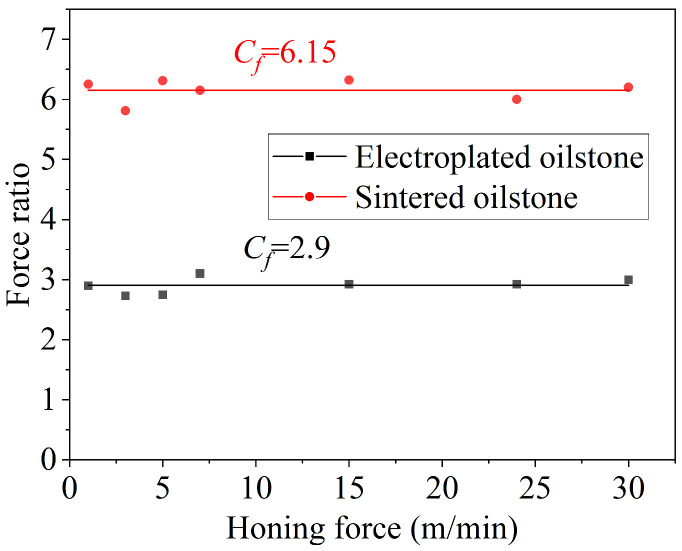
Influence of honing speed on force ratio.

**Figure 18 materials-17-06170-f018:**
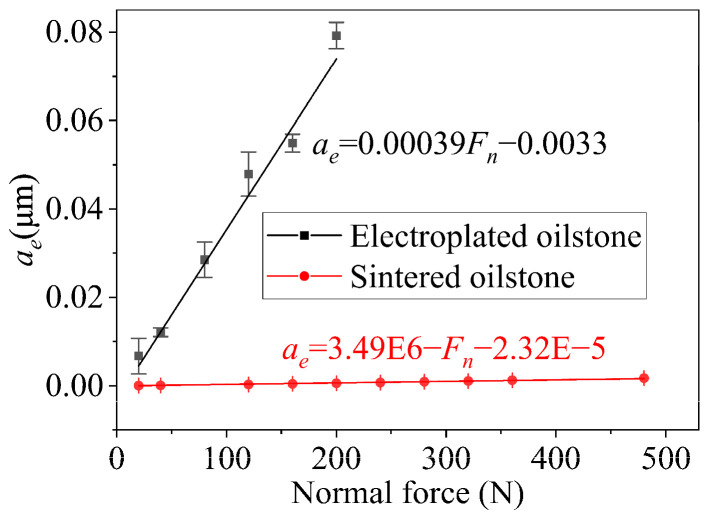
Influence of normal force on *a_e_*.

**Figure 19 materials-17-06170-f019:**
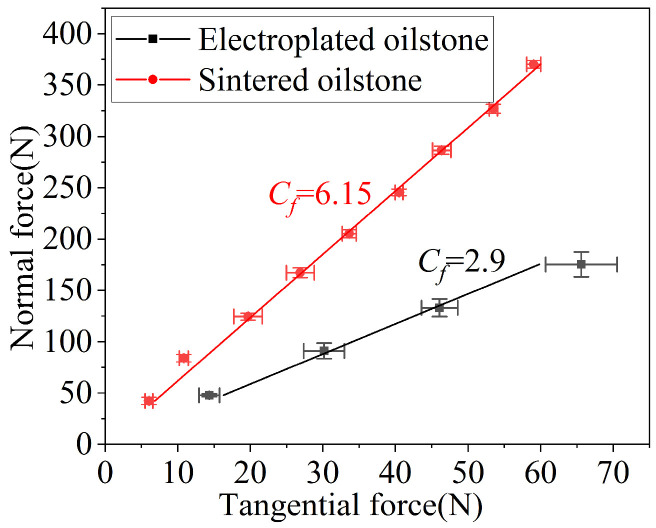
Influence of normal force on force ratio.

**Figure 20 materials-17-06170-f020:**
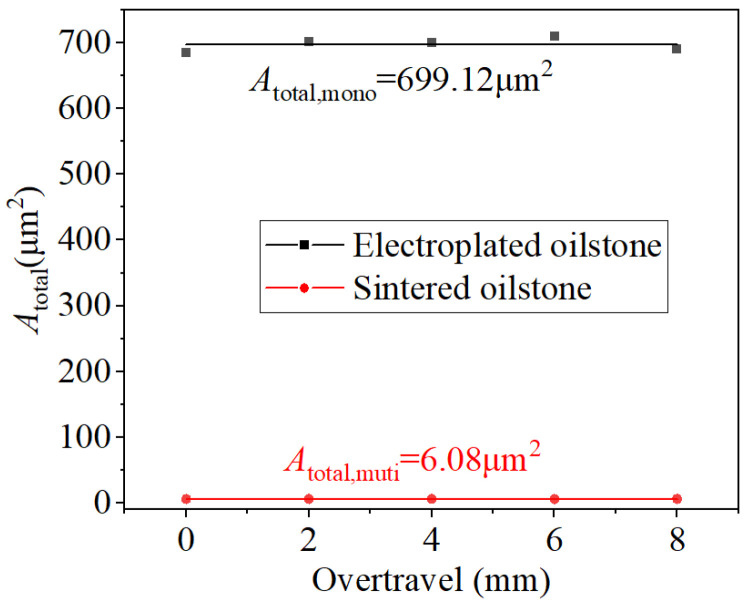
Influence of overtravel on *A*_total_.

**Figure 21 materials-17-06170-f021:**
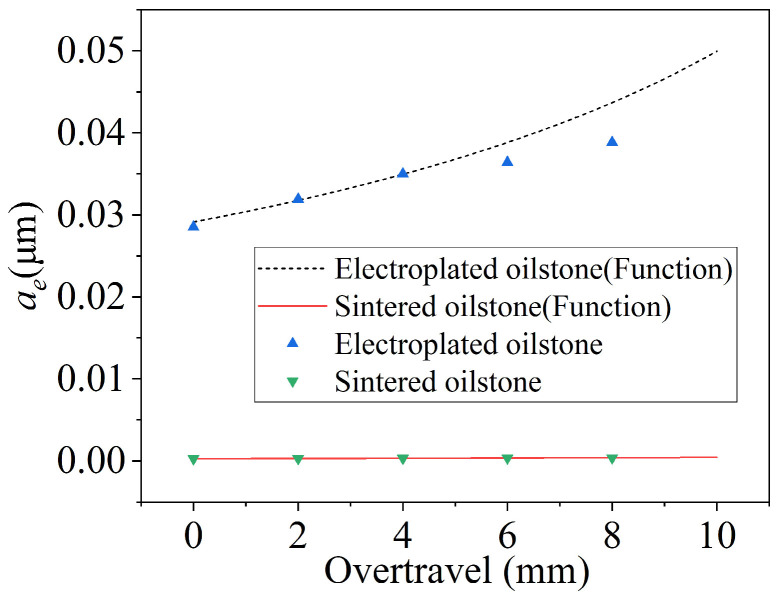
Influence of overtravel on *a_e_*.

**Figure 22 materials-17-06170-f022:**
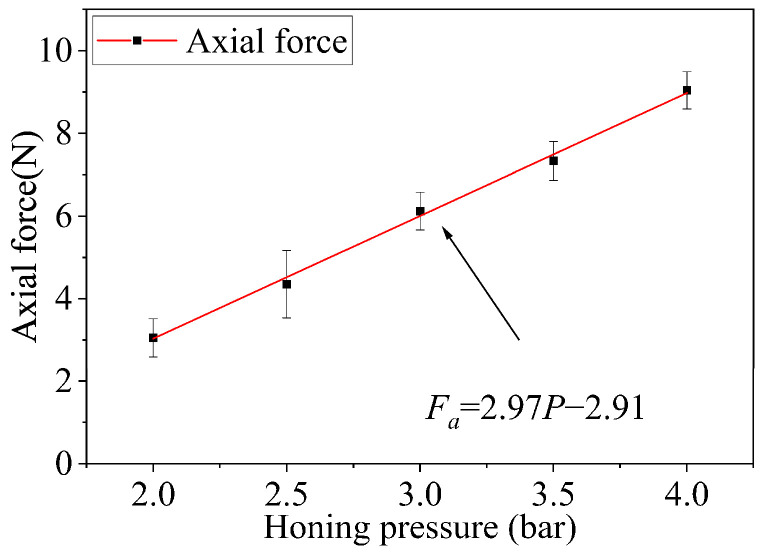
Influence of honing pressure on axial force (θ = 22.6°).

**Figure 23 materials-17-06170-f023:**
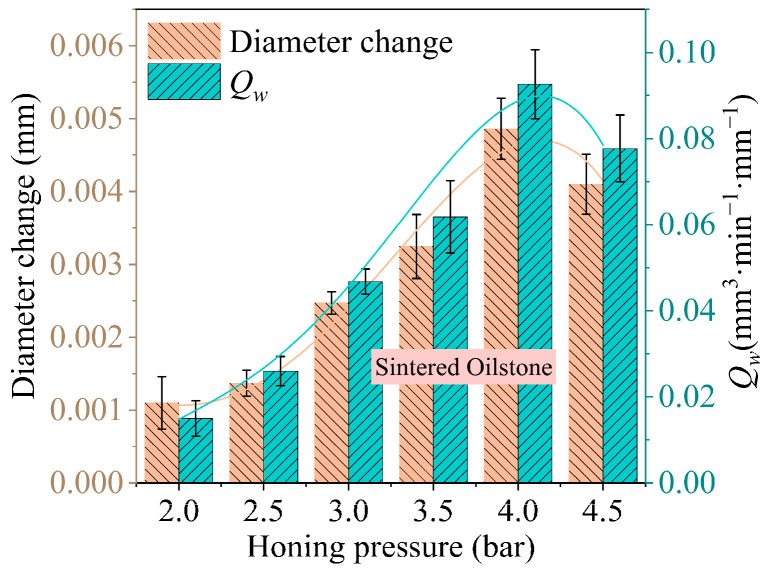
The impact of honing pressure on *Q_w_* with sintered oilstone.

**Figure 24 materials-17-06170-f024:**
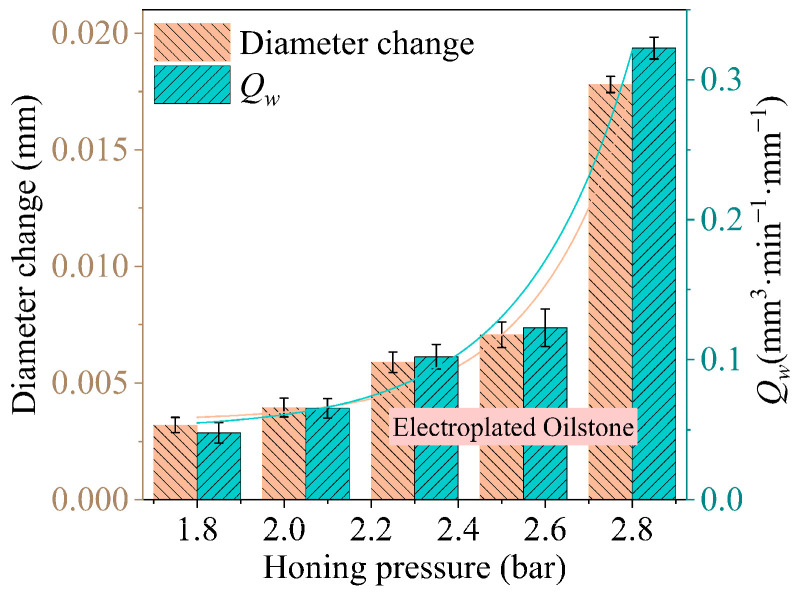
The impact of honing pressure on *Q_w_* with electroplated oilstone.

**Figure 25 materials-17-06170-f025:**
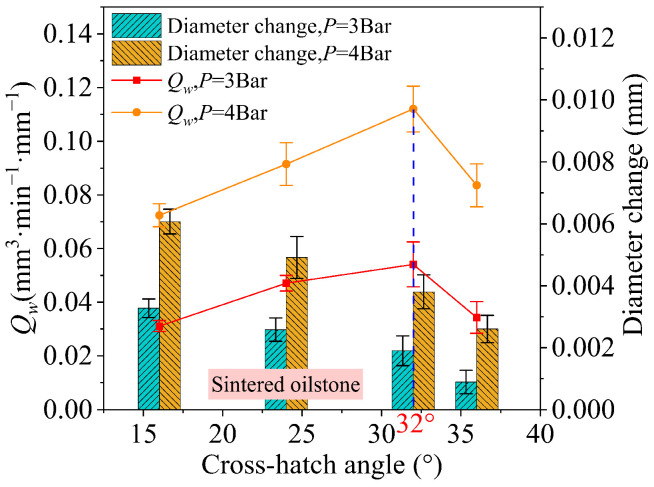
The impact of cross-hatch angle on *Q_w_* with sintered oilstone.

**Figure 26 materials-17-06170-f026:**
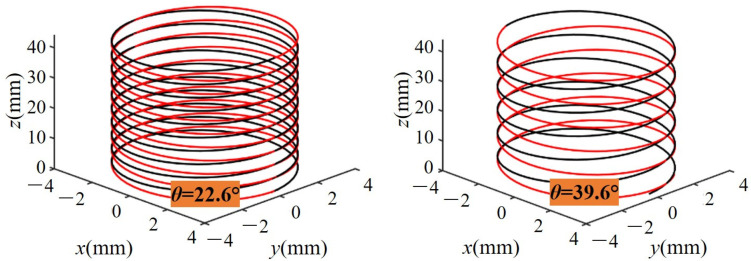
Honing trace at different cross-hatch angles.

**Figure 27 materials-17-06170-f027:**
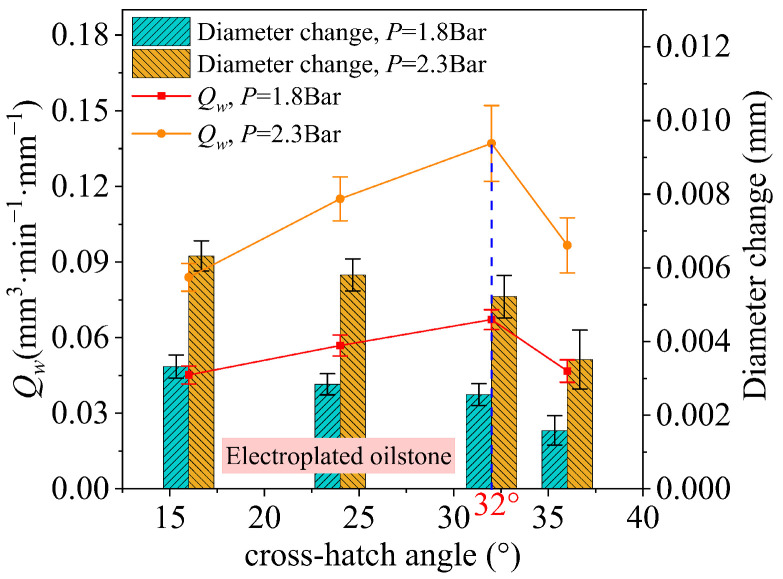
The impact of cross-hatch angle on *Q_w_* with electroplated oilstone.

**Figure 28 materials-17-06170-f028:**
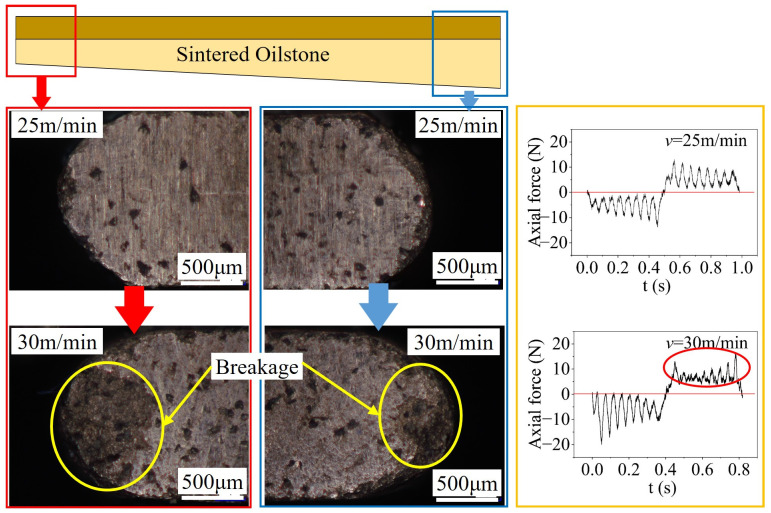
Fracture morphology of sintered oilstone.

**Figure 29 materials-17-06170-f029:**
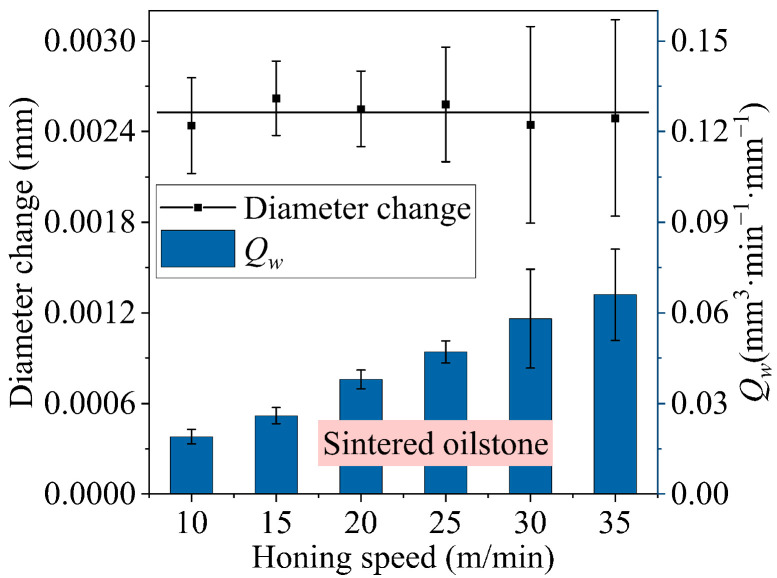
Influence of honing speed on *Q_w_* (sintered oilstone).

**Figure 30 materials-17-06170-f030:**
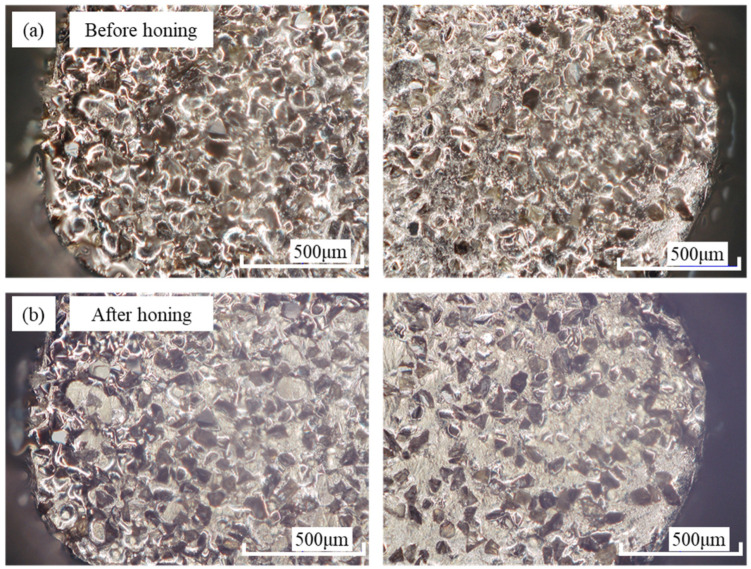
Comparison before and after honing with electroplated oilstone ((**a**) Before honing, (**b**) After honing).

**Figure 31 materials-17-06170-f031:**
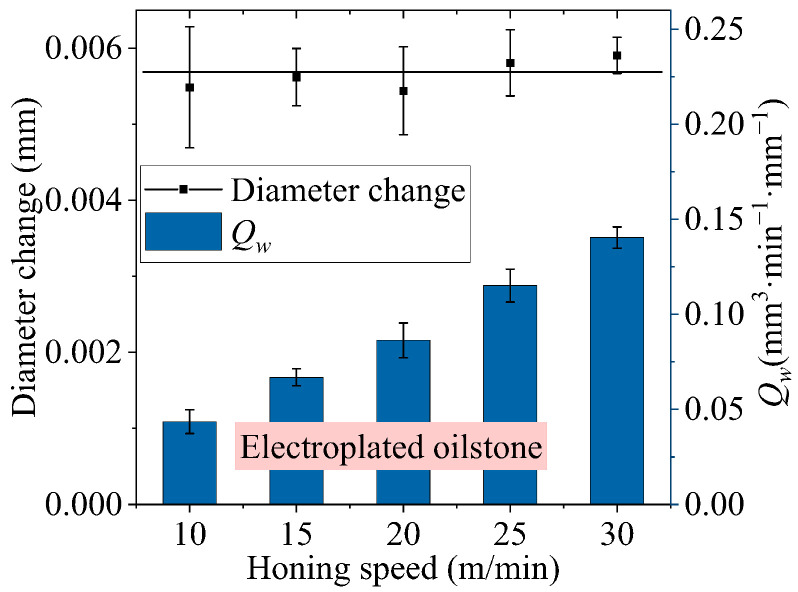
Influence of honing speed on *Q_w_* (electroplated oilstone).

**Figure 32 materials-17-06170-f032:**
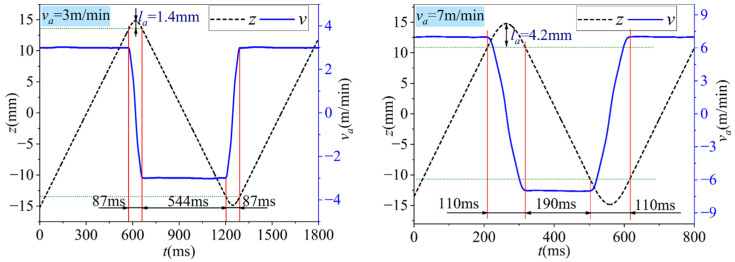
Oilstone position and real-time speed at different reciprocating speed.

**Figure 33 materials-17-06170-f033:**
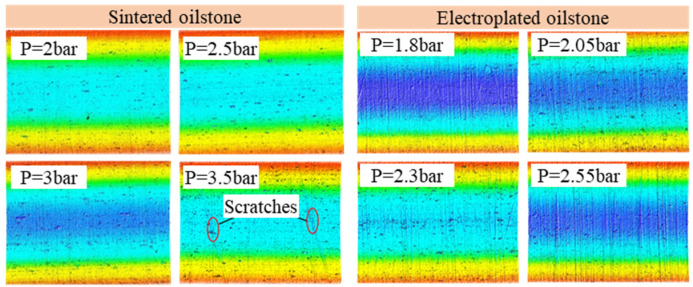
Surface topography of valve sleeve.

**Figure 34 materials-17-06170-f034:**
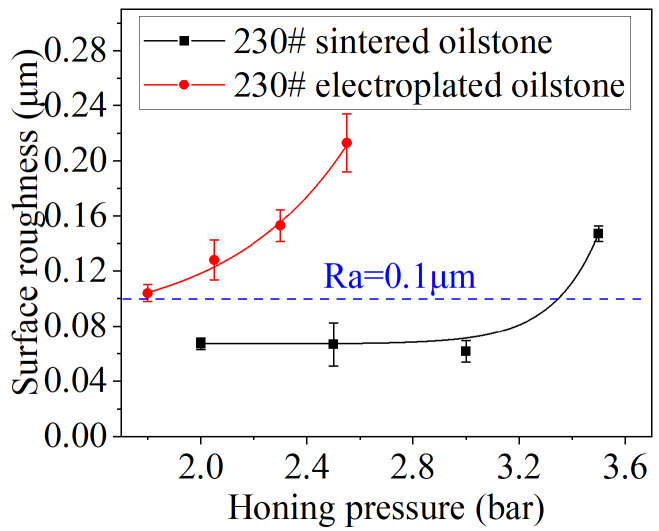
Influence of honing pressure on roughness.

**Figure 35 materials-17-06170-f035:**
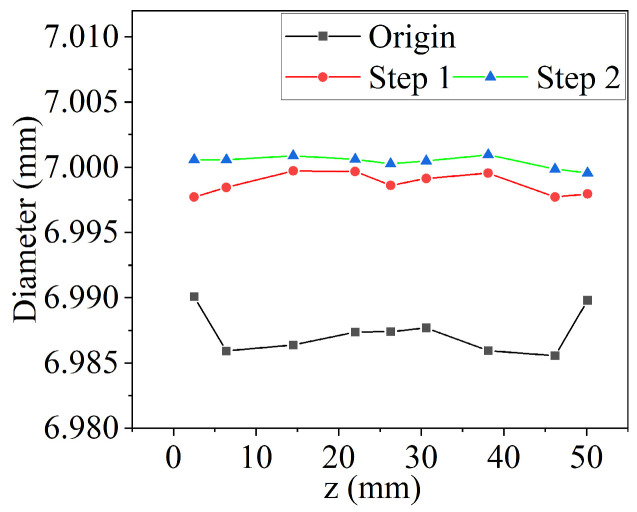
Change of valve sleeve diameter under process indicated in Table 5.

**Table 1 materials-17-06170-t001:** 9Cr18MoV composition [20].

Elements	C	Mn	Si	Cr	Mo	V	S	P
wt, %	0.85~0.95	<0.8	<0.8	17~19	1~1.3	0.07~0.12	<0.03	<0.035

**Table 2 materials-17-06170-t002:** Single-factor scratch test parameters.

Parameters	Types	
	Sintered Oilstone	Electroplated Oilstone
Speed (m/min)	1~30	1~30
Normal force (N)	40~480	20~200
Overtravel (mm)	0, 2, 4, 6, 8	0, 2, 4, 6, 8
Grit size	230#	230#
Stroke (mm)	18	18

**Table 3 materials-17-06170-t003:** Honing parameters.

Parameters	Sintered Oilstone	Electroplated Oilstone
Overtravel (mm)	2	2
Pressure (bar)	2, 2.5, 3, 3.5, 4, 4.5	1.8, 2.05, 2.3, 2.55, 2.8
Cross-hatch angle (°)	16, 24, 32, 36	16, 24, 32, 36
Hoing speed (m/min)	10, 15, 20, 25, 30, 35	10, 15, 20, 25, 30, 35

**Table 4 materials-17-06170-t004:** Test parameters of workpiece topography analysis.

Parameters	Sintered Oilstone	Electroplated Oilstone
Honing pressure (bar)	2, 2.5, 3, 3.5	1.8, 2.05, 2.3, 2.55
Hoing speed (m/min)	25	30
Cross-hatch angle (°)	32	

**Table 5 materials-17-06170-t005:** Valve sleeve processing steps and parameters.

Features	Step 1	Step 2
Oilstone type	230# electroplated oilstone	230# sintered oilstone
Allowance (mm)	about 0.012	about 0.002
Honing speed (m/min)	30	25
Strokes	120	30
Cross-hatch angle (°)	32	
Overtravel (mm)	8	

**Table 6 materials-17-06170-t006:** Process comparison between traditional process and that indicated in Table 5.

Process	Cylindricity (mm)	Roughness (μm)	Honing Time (s)	Correcting Time (s)	Total Time (s)
Process 1	0.0041	0.066	233	31	264
Process 2	0.0027	0.063	101	0	101

## Data Availability

The original contributions presented in the study are included in the article. Further inquiries can be directed to the corresponding authors.

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
