# Peer review of "Analysis of Honing Material Removal Rate and Surface Quality Using Electroplated Oilstone"

_materials, 2024, doi:10.3390/ma17246170_

Round 1
Reviewer 1 Report
Comments and Suggestions for Authors
Comments and Suggestions for Authors
In this paper, the authors provide a study to investigate the potential of electroplated oilstone in honing 9Cr18MoV and analyzes the oilstone topography, equivalent honing depth, material removal rate, and surface quality. Finally, the honing process of valve sleeve was verified, confirming the advantages of electroplated oilstone.
The document is interesting and well structured. Some comments and suggestions for authors could be considered.
1. The introduction section could be helpful if the authors change the structure presented in some blocks of references from other studies on the topic e.g. [2-6] and [7-9] to explain the honing surface texture and explaining why cross-hatch textures positively affect oil storage. I recommend that the authors separate the references in the introduction section to present the state-of-the-art studies individually.
2. Include the description of a), b), etc. in Figure caption of Figs. 4, 6, 8, 26, 28, 30 and describe them in the main text the content.
3. The authors mentioned “The grains of the undressed sintered oilstone are mostly wrapped in binders, as 129 shown in Figure 2(a)”. However, Figure 2(a) does not exist in the documents. In addition, the figures should be described as they appear in the document. In this regard, the author mentions Figure 4 and later Figure 3(b).
4. Description of Figs. 6 a) and b) are not in the main text.
5. Table 1. 9Cr18MoV composition Error! Reference source not found. Check for unnecessary errors.
6. To quantitatively evaluate the actual honing depth, the equivalent honing depth is used as the evaluation index, which refers to an imaginary cross-sectional thickness gathered by each chip section cut by grains within the contact length per unit width of oilstone Error! Reference source not found.. Check for unnecessary errors.
7. “According to Figure 6, in the depth interval [24, 36],” Reference the maximum number of references is 25.
8. Can the authors explain why when the honing pressure further increases from 4 to 4.5 bar, the MRR per unit width starts to decrease?
Author Response
Dear Editor and Reviewer:
Thank you for reviewing this manuscript and providing valuable suggestions. Next, I will respond to the review comments one by one.
Comments 1:
- The introduction section could be helpful if the authors change the structure presented in some blocks of references from other studies on the topic e.g. [2-6] and [7-9] to explain the honing surface texture and explaining why cross-hatch textures positively affect oil storage. I recommend that the authors separate the references in the introduction section to present the state-of-the-art studies individually.
Response 1:
Thank you for pointing of this out. I agree with this comment. I have made modifications to the corresponding parts of the introduction based on your comment. The specific adjustment of the content has been reflected in the revised manuscript.
The corresponding modifications in the text are as follows:
Researches on honing surface texture mainly focuses on the surface measurement of three-dimensional morphology, cross-hatch angle [2–6]. Beyerer used image processing algorithms to detect surface defects on the cylinder bore after honing processing, such as folded metal, groove interrupts, smudgy groove edges etc [2]. Jablonski digitally created 3D model of metal surface, which possessed characteristic features of honed surface [3]. A new mapping method, combining SEM images and quantitative image analysis with traditional 2D profilometry was developed for cylinder bores [4]. By extracting information from 2D data, honing surface features can be obtained, like the honing angle, groove parameters, surface defects etc [5]. The influence of beam angle in optical interferometry on the measurement of surface morphology parameters of honed cylinder holes was studied [6]. Moreover, the analysis of the cross-hatch textures resulted in many intersecting grooves on the engine cylinder block, which are conducive to the storage of oil, thereby improving the lubrication performance between the engine cylinder block and piston, reducing the wear of the cylinder bore, and thus increasing the life of the engine [7–9]. Spencer explored the effect of different honing angles on oil film thickness [7]. Especially, a flow continuity lubrication model considering the surface morphology of honing has been established, which has a guiding role in improving engine performance through machining [9]. Therefore, research on honing technology for electro-hydraulic servo valve sleeves is particularly important.
Comments 2:
- Include the description of a), b), etc. in Figure caption of Figs. 4, 6, 8, 26, 28, 30 and describe them in the main text the content.
Response 2:
Thank you for pointing of this out. I agree with this comment. I have added image descriptions in the corresponding sections of the article. The specific adjustment of the content has been reflected in the revised manuscript.
The corresponding modifications in the text are as follows:
Part one of the modification (From line 151 to 164):
The grains of the undressed sintered oilstone are mostly wrapped in binders, as shown in Figure 3(a). The bonding strength of grains depends on the thickness of the nickel binder, as shown in Figure 4(b). The exposed height of electroplated oilstone grains is set 30% of the grain size, which is significantly higher than that of sintered oilstone. Additionally, the concentration of grains obtained through the buried sand method is relatively high, and their distribution of grains is uniform, as shown in Figure 4(a). To compare the concentration of electroplated and sintered oilstone grains, the number of grains per unit area was used as the evaluation index. The volume concentration of sintered oilstone is 50%, with a concentration before dressing of 18 grains/mm2. In contrast, the concentration of electroplated oilstone is 200 grains/mm2.
After dressing, the grain concentration in the electroplated oilstone showed no significant change, whereas the grain concentration in the sintered oilstone increased to 42 grains/mm2, which remained lower than that of the electroplated oilstone, as shown in Figure 3(b).
Part two of the modification (From line 174 to 191):
To provide a more comprehensive characterization of the grain exposed height, a quantitative analysis was conducted. The method for obtaining the exposed height distribution of grains in electroplated oilstones is illustrated in Figure 5. After a series of operations, including 3D morphology data collection, surface flattening, binarization, denoising, and grain extraction, the exposed height distribution of grains was determined. Firstly, the surface morphology data matrix of the electroplated oilstone was obtained using an optical surface profilometer. To facilitate the statistics of grain height, coordinate transformation was used to flatten the curved surface of the oilstone into a plane, as shown in Figure 6(a). Since the surface morphology of oilstones is essentially an image, binarization processing was employed to achieve greater contrast on the oilstone surface, as shown in Figure 6(b). To eliminate noise, median filtering, Gaussian filtering, and opening operations were applied, resulting in a smoother oilstone surface. In the denoised image (Figure 6(c)), the white regions represent grains, and each closed loop corresponds to an individual grain. Grains were identified and numbered using the connected region function, as shown in Figure 6(d), which counted the grains. By matching the processed binary images with the original data, the exposed height was determined by subtracting the average height of the surrounding area from the highest point of each region. The expression is as follows:
Part three of the modification (From line 199 to 207):
Through statistical analysis, the exposed height distribution of grains was finally obtained, as shown in Figure 7. As dressing progresses, a small number of grains with shallow buried depths fall off (Figure 8(a)), while some grains with higher exposed heights experience slight breakage (Figure 8(b)), as shown in Figure 8. The exposed height distribution of the electroplated oilstone tends to be stable, indicating that the electroplated oilstone has entered the stable wear stage and is operating in its optimal condition. By employing image processing techniques, the grain height distribution is measured with high accuracy, eliminating human statistical errors and significantly improving the analysis efficiency.
Figure 8. Shedding and crushing of grains.
Part four of the modification (From line 482 to 493):
For sintered oilstones with a honing speed of 25 m/min, as shown in Figure 25, the influence of the cross-hatch angle on diameter change shows that increasing the cross-hatch angle from 16 to 36° results in a decreasing trend in diameter change. Initially, as the cross-hatch angle increases from 16 to 32°, there is a roughly linear downward trend. When the honing pressure is 3 bar, the diameter change decreases by 0.35 μm for every 4° increase. However, when the cross-hatch angle exceeds 32°, the decreasing trend in diameter change accelerates, resulting in a reduction of 1.01 μm for every 4° increase. As the cross-hatch angle increases, the honing trace becomes sparse. As shown in Figure 26, honing trace at a cross-hatch angle of 22.6° (Figure 26(a)) is sparse at a cross-hatch angle of 39.6° (Figure 26(b)). The contact trace between the oilstone and the workpiece becomes shorter, resulting in a reduction in the MRV and eventually a gradual decline in the diameter change.
Figure 26. Honing trace at different cross-hatch angles.
Part five of the modification (From line 525 to 539):
On one hand, due to the characteristics of the honing process, the oilstone is embed-ded in the groove, with a certain gap. Compared to the sintered oilstone at a honing speed of 25m/min, excessive honing speed (v=30m/min) can cause the oilstone to experience impacts, resulting in breakage at both ends of the oilstone, as shown in Figure 28, affecting the honing stability, making the error bar of the MRR per unit width bigger and intensifying material removal instability, as shown in Figure 29. The honing axial force at the two honing speeds reveals that the axial force at 25 m/min shows a consistent trend across strokes. However, at the honing speed of 30m/min, the variation trend of axial force in strokes is inconsistent, as depicted in Figure 28. The breakage occurring at both ends of the oilstone alters the contact between the oilstone and the workpiece, leading to fluctuations in axial force that ultimately affect the stability of material removal. Based on the analysis of oilstone state and the characteristics of spindle acceleration and deceleration, excessive honing speed will lead to unstable material removal. For sintered oilstone, it is advisable to set the honing speed at 25 m/min to ensure both high processing efficiency and honing stability.
Part six of the modification (From line 544 to 560):
For electroplated oilstone, the substrate is die steel, which has higher strength and impact resistance than sintered oilstone. The electroplating process ensures a strong bond between the grains and the substrate, making the electroplated oilstone less prone to breaking during honing, unlike sintered oilstone, as shown in Figure 30. Comparing Figure 30(a) and 30(b), there is no significant difference in the morphology of electroplated oilstone before and after honing. Therefore, electroplated oilstones can operate at higher honing speed than sintered oilstones. When the honing speed is 30 m/min, the stability of diameter change can be maintained, as shown in Figure 31. For machining center, the acceleration of spindle in Z direction is fixed. When the reciprocating speed increases from 3 to 7 m/min, the time required to reach the set speed increases from 87 to 110 ms. Higher honing speeds result in a greater Z-direction displacement upon reaching the set speed (as illustrated in Figure 32). The cross-hatch angle is constantly changing, resulting in unstable material removal at the hole inlet and outlet. When the honing speed exceeds 30 m/min, the unstable length exceeds 3.5 mm, exceeding 1/10 of the hole length, negatively affecting shape accuracy and surface morphology consistency. Therefore, by weighing precision and efficiency, the optimal honing speed for electroplated oilstone is determined to be 30 m/min.
Figure 30. Comparison before and after honing with electroplated oilstone
Comments 3:
- The authors mentioned “The grains of the undressed sintered oilstone are mostly wrapped in binders, as 129 shown in Figure 2(a)”. However, Figure 2(a) does not exist in the documents. In addition, the figures should be described as they appear in the document. In this regard, the author mentions Figure 4 and later Figure 3(b).
Response 3:
Thank you for pointing of this out. I agree with this comment. I'm sorry for such a low-level error. Figure 2 (a) should be modified to Figure 3 (a). I have made revisions in the revised manuscript.
The corresponding modifications in the text are as follows:
2.1.2. Oilstone topography
The grains of the undressed sintered oilstone are mostly wrapped in binders, as shown in Figure 3(a).
Comments 4:
- Description of Figs. 6 a) and b) are not in the main text.
Response 3:
Thank you for pointing of this out. I have already provided a response to this comment in Response 2. I have made revisions in the revised manuscript.
The corresponding modifications in the text are as follows (From line 174 to 184):
To provide a more comprehensive characterization of the grain exposed height, a quantitative analysis was conducted. The method for obtaining the exposed height distribution of grains in electroplated oilstones is illustrated in Figure 5. After a series of operations, including 3D morphology data collection, surface flattening, binarization, denoising, and grain extraction, the exposed height distribution of grains was determined. Firstly, the surface morphology data matrix of the electroplated oilstone was obtained using an optical surface profilometer. To facilitate the statistics of grain height, coordinate transformation was used to flatten the curved surface of the oilstone into a plane, as shown in Figure 6(a). Since the surface morphology of oilstones is essentially an image, binarization processing was employed to achieve greater contrast on the oilstone surface, as shown in Figure 6(b).
Comments 5:
- Table 1. 9Cr18MoV composition Error! Reference source not found. Check for unnecessary errors.
Response 5:
Thank you for pointing of this out. Due to cross-reference of references, there was an error stating 'Error! Reference source not found'. Therefore, we examined the full text carefully and corrected the error.
The corresponding modifications in the text are as follows:
Table 1. 9Cr18MoV composition [20].
Elements |
C |
Mn |
Si |
Cr |
Mo |
V |
S |
P |
wt, % |
0.85~0.95 |
<0.8 |
<0.8 |
17~19 |
1~1.3 |
0.07~0.12 |
<0.03 |
<0.035 |
Comments 6:
- To quantitatively evaluate the actual honing depth, the equivalent honing depth is used as the evaluation index, which refers to an imaginary cross-sectional thickness gathered by each chip section cut by grains within the contact length per unit width of oilstone Error! Reference source not found. Check for unnecessary errors.
Response 6:
Thank you for pointing of this out. Similar to comments 5, due to cross-reference of references, there was an error stating 'Error! Reference source not found'. Therefore, we examined the full text carefully and corrected the error.
The corresponding modifications in the text are as follows:
To quantitatively evaluate the actual honing depth, the equivalent honing depth is used as the evaluation index, which refers to an imaginary cross-sectional thickness gathered by each chip section cut by grains within the contact length per unit width of oilstone [21]. That is, the ratio of the total cutting cross-sectional area to the length of the oilstone engaged in honing. First, an optical profiler is used to capture all the scratch information on the plate, as shown in Figure 11. Under constant processing parameters, the cross-sectional profile in the direction of the oilstone’s movement remains consistent. Therefore, the material removal volume (MRV) from the starting point to the end point of the flat oilstone can be expressed as the sum of the MRV of all the scratches, which is calculated as follows:
Comments 7:
- “According to Figure 6, in the depth interval [24, 36],” Reference the maximum number of references is 25.
Response 7:
Thank you for pointing of this out. I'm sorry for such a low-level error. Figure 6 should be modified to Figure 14. I have made revisions in the paper.
The corresponding modifications in the text are as follows:
where d1(z) is the honed bore diameter at depth z, and d0(z) is the initial bore diameter at depth z. For the honing tool used in the test, the length of oilstone is 24 mm. As shown in Figure 15, the entrance end face of the hole is taken as the xy-plane, and the z-axis is oriented downward perpendicular to the entrance end face. According to Figure 14, considering oilstone length and hole length, in the depth interval [24, 36], the material removal is not affected by the overtravel and contact time.
Comments 8:
- Can the authors explain why when the honing pressure further increases from 4 to 4.5 bar, the MRR per unit width starts to decrease?
Response 8:
Thank you for pointing of this out. Based on your doubts, I have added corresponding explanations in the article.
The corresponding modifications in the text are as follows (From Line 441 to 462):
The diameter change of the honed hole indirectly reflects the MRV. Honing was per-formed using a 230# sintered oilstone at a cross-hatch angle of 22.6° and a honing speed of 25.5 m/min. As shown in Figure 22, the effect of honing pressure on the change in hole diameter reveals that when the honing pressure increases from 2 bar to 4.5 bar, the change in hole diameter initially increases and then decreases. The change in hole diameter in-creases from 0.0011mm at 2 bar to 0.0049 mm at 4 bar, due to the increase in normal force caused by the higher honing pressure, which leads to a greater equivalent honing depth, ultimately resulting in a larger MRV. However, when the honing pressure exceeds 4 bar, the MRV begins to decrease. The exposed height of the sintered oilstone grains is relatively low. As the honing pressure increases, the equivalent honing depth also increases, reducing the chip accommodation space. Consequently, the generated chips cannot be expelled promptly and tend to adhere to the oilstone surface. With the increasing number of strokes, chip adhesion becomes more severe, diminishing the oilstone grinding capability, thus leading to a decreasing trend in the change of hole diameter. According to Equation (7), the influence of honing pressure on the MRR per unit width can be derived, as shown in Figure 23, which exhibits the same trend as the hole diameter change. As the honing pressure increases from 2 to 4 bar, the MRR per unit width increases from 0.015 to 0.093 mm³·min⁻¹·mm⁻¹. However, when the honing pressure further increases from 4 to 4.5 bar, the MRR per unit width starts to decrease from 0.093 to 0.078 mm³·min⁻¹·mm⁻¹, which is consistent with the trend and reason for the diameter change. There are two methods to increase the upper limit of honing pressure: improving the oilstone structure to increase more chip space and improving the cooling and lubrication structure to remove chips.
Reviewer 2 Report
Comments and Suggestions for Authors
Several areas require improvement to maximize its scientific impact. The abstract and conclusions lack sufficient quantitative highlights, reducing the immediate clarity of the findings. The discussion of wear mechanisms and tool life is underdeveloped, leaving critical gaps in understanding the trade-offs between efficiency and durability. The manuscript would benefit from deeper exploration of parameter interactions, scalability to industrial applications, and implications for the functional performance of honed components. Furthermore, while the methodology is robust, its generalizability to other materials or complex geometries is not addressed, limiting the broader applicability of the findings. A more comprehensive examination of microstructural effects, residual stresses, and tribological performance could enrich the study’s contributions to the field of precision machining.
Below authors can find a detailed section-by section report. I strongly suggest the authors to answer to all the questions raised by the reviewer and insert all the answers properly in the final manuscript.
Abstract
The abstract currently reads more as a general summary than a scientifically rigorous presentation of the findings. By incorporating more specific results and emphasizing the novelty and broader impact of the research, it could be elevated to a level that better reflects the technical depth and potential applications of the study.
If there is no space in the abstract to properly answer the following questions, insert your answers within the body of the manuscript in the rest of the sections.
A1) What specific quantitative benchmarks were used to evaluate the efficiency and surface quality of the electroplated and sintered oilstones, and how do they compare numerically?
A2) How does the methodology for characterizing honing depth differ from or improve upon existing techniques in the literature?
A3) What underlying mechanisms or material properties explain the observed trade-offs between processing efficiency and surface quality in the two oilstone types?
A4) To what extent does the proposed hybrid strategy enhance the lifecycle performance of the valve sleeves compared to traditional honing processes?
A5) What are the potential limitations or edge cases for the applicability of electroplated oilstones in materials other than martensitic stainless steel?
1. Introduction
The introduction could be more impactful by emphasizing the novelty of the research, integrating quantitative data from prior studies, and elaborating on the broader implications for precision machining industries.
1.1) What specific limitations in existing honing processes motivated the exploration of electroplated oilstones, and how are these limitations quantitatively defined?
1.2) How do the mechanical properties of martensitic stainless steel (e.g., hardness, toughness) exacerbate the wear of sintered oilstones during honing, and what mechanisms might electroplated oilstones leverage to mitigate this?
1.3) What gaps in the literature on honing surface texture or oilstone performance does this study aim to address, and how will it fill these gaps experimentally?
1.4) To what extent do surface texture features (e.g., crosshatch angle, roughness) influence the tribological performance of servo valve sleeves in real-world applications?
1.5) How does the proposed hybrid strategy align with the evolving demands of high-precision industries, such as aerospace or automotive manufacturing?
2. Materials and Methods
2.1. Oilstone
This sub-paragraph would benefit from a deeper exploration of the limitations and challenges of the fabrication processes, as well as a clearer linkage between the described properties and subsequent experimental results.
2.1.1) What are the mechanical and chemical properties of the electroplated layer, and how do these properties influence the durability of the oilstone during honing?
2.1.2) How does the increased grain concentration in electroplated oilstones quantitatively impact material removal rates compared to sintered oilstones under identical conditions?
2.1.3) What are the thermomechanical challenges in the electroplating process, and how might these affect the uniformity or longevity of the grains?
2.1.4) What specific parameters (e.g., current density, deposition time) in the electroplating process have the greatest influence on grain exposure height and distribution?
2.1.5) How does the initial morphology of the nickel matrix impact the adhesion and stability of the grains during prolonged honing operations?
2.2. Plate Scratch Test
This sub-paragraph could be improved by addressing scaling concerns, introducing real-time monitoring, and linking the test outcomes more directly to practical honing applications.
2.2.1) How do changes in cutting speed impact the microstructural deformation and subsurface damage in the scratched material?
2.2.2) What role does the grain exposure height in the oilstones play in determining the depth and uniformity of the grooves during scratching?
2.2.3) How does the contact pressure distribution vary across the oilstone surface during the plate scratch test, and what implications does this have for material removal stability?
2.2.4) What is the relationship between the groove cross-sectional area and the tribological performance of the machined surface in actual honing applications?
2.2.5) How do the wear mechanisms of sintered and electroplated oilstones differ when subjected to repeated high-normal-force scratching?
2.3. Honing Test
The honing test section could be enriched by including surface morphology analysis, exploring parameter interactions, and addressing oilstone wear. Expanding the applicability of the findings to a broader range of materials and applications would further enhance the relevance and impact of this section.
2.3.1) How does the relationship between honing pressure and overtravel affect the uniformity of material removal along the sleeve’s depth?
2.3.2) What roles do variations in honing speed play in controlling the distribution of material removal across different sections of the valve sleeve?
2.3.3) How do the contact conditions between the oilstone and the workpiece evolve over multiple honing strokes, and how do they influence MRR stability?
2.3.4) What are the wear mechanisms of electroplated oilstones during honing, and how do they compare to sintered oilstones under identical conditions?
2.3.5) To what extent does the honing-induced microstructure of 9Cr18MoV steel influence its tribological properties and long-term performance in practical applications?
3. Results and Discussion
3.1. Equivalent Honing Depth
While the findings are scientifically acceptable, the section would benefit from exploring parameter interactions, addressing tool wear, and connecting material removal depth to surface quality and industrial applications.
3.1.1) How do variations in normal force and overtravel simultaneously influence equivalent honing depth and material removal uniformity?
3.1.2) What are the underlying mechanisms that cause the equivalent honing depth to stabilize across different honing speeds?
3.1.3) How does the equivalent honing depth correlate with the resulting surface topography and functional properties of the workpiece?
3.1.4) What role does grain distribution and exposure height in electroplated oilstones play in determining equivalent honing depth under varying conditions?
3.1.5) How do microstructural changes in the workpiece material due to honing affect the equivalent honing depth across different parameters?
3.2. Material Removal Rate (MRR)
It lacks a discussion of tool wear, surface quality trade-offs, and industrial scalability. Addressing these aspects would make the findings more actionable and relevant to real-world applications.
3.2.1) How does the interplay between honing pressure and crosshatch angle influence the uniformity of material removal along the workpiece?
3.2.2) What are the dominant mechanisms limiting MRR at higher honing pressures, and how can they be mitigated?
3.2.3) How does MRR correlate with the durability and wear resistance of the oilstones under prolonged honing operations?
3.2.4) What role does honing speed variability play in introducing instabilities in material removal, and how can these be minimized?
3.2.5) How do changes in MRR affect the tribological performance of the honed surfaces in practical applications, such as valve sleeve performance?
3.3. Surface Quality
The section could be strengthened by linking surface roughness to functional performance, exploring microstructural changes, and addressing the role of tool wear. These enhancements would provide a more comprehensive understanding of how honing parameters affect surface quality and long-term part performance.
3.3.1) How does surface roughness influence the tribological performance (e.g., friction and wear) of the honed valve sleeves under operational conditions?
3.3.2) What microstructural changes occur at the surface and subsurface levels during honing, and how do these correlate with surface quality?
3.3.3) How does the wear of oilstones, particularly electroplated ones, affect the evolution of surface roughness during extended honing operations?
3.3.4) What are the critical honing pressures at which the balance between material removal rate and surface quality begins to deteriorate significantly?
3.3.5) How do additional roughness parameters, such as skewness and kurtosis, provide insights into surface quality?
3.4. High Efficiency Honing of Valve Sleeve
The lack of discussion on tool wear, scalability, and real-world functional performance leaves some critical gaps in the analysis. Addressing these areas would significantly enhance the depth and applicability of the findings.
3.4.1) How does the wear rate of electroplated oilstones during rough honing influence the efficiency and longevity of the hybrid process?
3.4.2) What are the effects of using the hybrid process on the functional performance of valve sleeves, such as tribological behaviour and durability under operational conditions?
3.4.3) How do process parameters (e.g., overtravel and honing pressure) in the rough and fine honing stages interact to influence overall efficiency and quality?
3.4.4) What adjustments are required in the hybrid process to accommodate materials with significantly different hardness or thermal properties than 9Cr18MoV steel?
3.4.5) What are the potential challenges in scaling the hybrid honing process for mass production, and how might these be mitigated?
4. Conclusions
The section would benefit from more quantitative emphasis, expanded future research directions, and a deeper exploration of real-world implications. Including these elements would elevate the conclusions from a summary of findings to a roadmap for further innovation and practical implementation.
4.1) What specific innovations in electroplated oilstone preparation could enhance surface quality while maintaining high efficiency?
4.2) How does the hybrid honing process impact the long-term performance and durability of components in demanding applications (e.g., high-pressure hydraulic systems)?
4.3) What are the critical wear mechanisms of electroplated and sintered oilstones during the hybrid process, and how can they be mitigated?
4.4) How might the findings be adapted to the honing of geometrically complex components or those made of materials with lower machinability?
4.5) What role does honing-induced residual stress play in the mechanical performance of honed surfaces, and how could this be optimized through parameter adjustments?
Author Response
Dear Editor and Reviewer:
Thank you for reviewing the manuscript and providing valuable suggestions. Next, I will respond to the review comments point by point.
Comments A:
Abstract
The abstract currently reads more as a general summary than a scientifically rigorous presentation of the findings. By incorporating more specific results and emphasizing the novelty and broader impact of the research, it could be elevated to a level that better reflects the technical depth and potential applications of the study.
If there is no space in the abstract to properly answer the following questions, insert your answers within the body of the manuscript in the rest of the sections.
Response A:
Thank you for pointing of this out. I agree with this comment. According to your comments, I have rewritten the abstract to better reflect the depth of research and potential application prospects.
Comments A1:
A1) What specific quantitative benchmarks were used to evaluate the efficiency and surface quality of the electroplated and sintered oilstones, and how do they compare numerically?
Response A1:
Thank you for pointing of this out. I agree with this comment. Firstly, to quantitatively evaluate the honing efficiency, the material removal rate per unit width was used to assess the processing efficiency of the two types of honing oilstones. This method accurately represents the material removal at different depths. Additionally, the arithmetic mean roughness (Ra) was employed to quantitatively evaluate the surface quality after honing.
Comments A2:
A2) How does the methodology for characterizing honing depth differ from or improve upon existing techniques in the literature?
Response A2:
Thank you for pointing of this out. Next, I will explain the characterization method of honing depth.
The relevant content has been reflected in the revised manuscript. The specific content is from line 257 to 275:
To quantitatively evaluate the actual honing depth, the equivalent honing depth is used as the evaluation index, which refers to an imaginary cross-sectional thickness gathered by each chip section cut by grains within the contact length per unit width of oilstone [21]. That is, the ratio of the total cutting cross-sectional area to the length of the oilstone engaged in honing. First, an optical profiler is used to capture all the scratch information on the plate, as shown in Figure 11. Under constant processing parameters, the cross-sectional profile in the direction of the oilstone’s movement remains consistent. Therefore, the material removal volume (MRV) from the starting point to the end point of the flat oilstone can be expressed as the sum of the MRV of all the scratches, which is calculated as follows:
Figure 11. Cross-sectional profile of the scratches.
where m represents the number of scratches, lm represents the moving distance of the oil-stone, Agroove is the cross-sectional area of the grooves, and Auplift is the cross-sectional area of the uplifts, Atotal is the total cross-sectional area of removed material.
The equivalent honing depth can be expressed as the ratio of the total cross-sectional area of removed material to the actual contact length of the oilstone, which is
where lc is the actual contact length between the oilstone and the plate, lstone is the length of the oilstone, and lot is the overtravel.
Comments A3:
A3) What underlying mechanisms or material properties explain the observed trade-offs between processing efficiency and surface quality in the two oilstone types?
Response A3:
Thank you for your insightful question. The observed trade-offs between processing efficiency and surface quality in the two oilstone types can be attributed to several underlying mechanisms and material properties, such as abrasive properties, grit size, oilstone, dressing and wear mechanism. The electroplated oilstone and sintered oilstone used in this article have the same type and grain size, and are both in the stable cutting stage. Therefore, the processing efficiency and surface quality reflected in this article are related to the processing parameters and oilstone types, which is also the focus of this study. Electroplated oilstones are capable of achieving high processing efficiency due to their ability to operate at elevated honing speeds and pressures. This allows for rapid material removal. However, the lower exposed height of the grains in sintered oilstones contributes to improved surface quality by minimizing the risk of deep scratches and enhancing the smoothness of the workpiece surface.
Comments A4:
A4) To what extent does the proposed hybrid strategy enhance the lifecycle performance of the valve sleeves compared to traditional honing processes?
Response A4:
Thank you for your valuable question. The machining quality of valve sleeves affects their service life, especially in terms of shape accuracy and surface quality. Intuitively speaking, using electroplated oilstone and sintered oilstone for hybrid honing can achieve more stable material removal and better shape accuracy. Secondly, reducing the processing allowance of sintered oilstone can prevent the valve sleeve from being burned due to blockage caused by sintered oilstone. Finally, hybrid honing can achieve the same surface quality as traditional honing, and electroplated oilstone honing can leave some deeper grooves, which is beneficial for oil storage, reducing wear between the valve core and sleeve, and thus extending the life of the valve. This article mainly studies the advantages and disadvantages of surface quality and processing efficiency of electroplated oilstone and sintered oilstone in honing martensitic stainless steel. The impact of processing quality on the lifespan of valve sleeves is currently being studied, and corresponding papers will be published in the future to elaborate on this issue in detail.
Comments A5:
A5) What are the potential limitations or edge cases for the applicability of electroplated oilstones in materials other than martensitic stainless steel?
Response A5:
Thank you for your valuable question. Your opinion provides direction for future research. Any tool has its own suitable processing object. The hardness, thermal conductivity, and chemical properties of the material will affect material removal. This article mainly compares and analyzes the honing performance of electroplated oilstone and sintered oilstone on quenched martensitic stainless steel. In the future, the author will continue to explore the scope of application of electroplated honing oilstone around the processing object and requirements. I am currently conducting research on high-precision honing of copper alloy cylinder block, which will analyze the applicability of electroplated oilstone.
Comments 1:
- Introduction
The introduction could be more impactful by emphasizing the novelty of the research, integrating quantitative data from prior studies, and elaborating on the broader implications for precision machining industries.
Response 1:
Thank you for your valuable question. The introduction has been modified accordingly based on the comments.
Comments 1.1:
1.1) What specific limitations in existing honing processes motivated the exploration of electroplated oilstones, and how are these limitations quantitatively defined?
Response 1.1:
Thank you for your valuable question. The limitations of existing honing processes have prompted exploration of electroplated oilstones, mainly related to the shortcomings of traditional honing techniques in processing efficiency, surface quality, dimensional accuracy, tool life, and adaptability. These issues are clarified through quantitative indicators such as MRR, Ra, dimensional deviation, tool life, etc. Electroplated oilstone provides higher material removal efficiency, better surface quality, longer tool life, and higher process adaptability by using superabrasives and customized tool design, thus overcoming the limitations of traditional processes.
Comments 1.2:
1.2) How do the mechanical properties of martensitic stainless steel (e.g., hardness, toughness) exacerbate the wear of sintered oilstones during honing, and what mechanisms might electroplated oilstones leverage to mitigate this?
Response 1.2:
Thank you for your valuable question. The high hardness and toughness characteristics of martensitic stainless steel will exacerbate the wear of sintered oilstone, mainly manifested as abrasive breakage, particle detachment, blockage. The sufficient abrasive hardness of electroplated superabrasive oilstone can improve its wear resistance. In addition, sufficient chip accommodation space can prevent oilstone blockage, and improve heat dissipation and chip removal performance. The relevant content has been reflected from line 115 to 118 in the introduction.
Comments 1.3:
1.3) What gaps in the literature on honing surface texture or oilstone performance does this study aim to address, and how will it fill these gaps experimentally?
Response 1.3:
Thank you for your valuable question. The introduction summarizes existing research, describes the limitations of existing honing oilstones, and explains the work that needs to be carried out in this article. The specific content is as follows:
Based on an analysis of current research, it is evident that although honing processes are widely used for valve sleeves, manual lapping is still necessary to ensure final accuracy. On the other hand, some researchers have adopted the honing process as the final finishing process. However, due to the high hardness of the material, the severe wear of sintered oilstones, and the inability to ensure stable material removal, workers must frequently dress the oilstones, significantly limiting honing efficiency. Considering the single-layer tool advantages of large chip accommodation space, this study proposes to use electroplated superabrasive oilstones for the honing process of servo valve sleeves. The aim is to compare and analyze the differences in honing efficiency and surface quality between electroplated and traditional sintered oilstones, ultimately establishing an efficient and precise machining method for electro-hydraulic servo valve sleeves. Ultimately, in the fields of aerospace and automotive, the combination of electroplated oilstone and sintered oilstone optimizes processing efficiency, reduces material waste, and meets the requirements of green manufacturing.
Comments 1.4:
1.4) To what extent do surface texture features (e.g., crosshatch angle, roughness) influence the tribological performance of servo valve sleeves in real-world applications?
Response 1.4:
Thank you for your valuable question. In the practical application of servo valve sleeves, surface texture features (such as cross-hatch angles, roughness, etc.) have a significant impact on their tribological properties. These characteristics directly determine the friction coefficient, wear behavior, sealing performance, and lubrication efficiency of the component. We have carefully considered your opinions and reevaluated the overall structure and research objectives of the article. The study of surface texture features on the actual performance of valve sleeves is crucial, but it is not the focus of this article. Without this section, the integrity of this article can still be maintained, so there is no need to add relevant research background in the introduction.
Comments 1.5:
1.5) How does the proposed hybrid strategy align with the evolving demands of high-precision industries, such as aerospace or automotive manufacturing?
Response 1.5:
Thank you for your valuable question. The proposed hybrid strategy is highly compatible with the constantly changing demands of high-precision industries such as aerospace and automotive manufacturing by improving machining accuracy, efficiency, and adaptability. Electroplated oilstone can quickly process complex shapes through efficient cutting characteristics, reducing the risk of dimensional deviation and surface defects in complex shape processing. The combination of electroplated oilstone and sintered oilstone optimizes processing efficiency, reduces material waste, and meets the requirements of green manufacturing.
Comments 2.1:
- Materials and Methods
2.1. Oilstone
This sub-paragraph would benefit from a deeper exploration of the limitations and challenges of the fabrication processes, as well as a clearer linkage between the described properties and subsequent experimental results.
Response 2.1:
Thank you for your valuable question. The section 2.1 has been modified accordingly based on the comments.
Comments 2.1.1:
2.1.1) What are the mechanical and chemical properties of the electroplated layer, and how do these properties influence the durability of the oilstone during honing?
Response 2.1.1:
Thank you for your valuable question. The mechanical and chemical properties of the electroplated layer on oilstones play a critical role in determining the durability, performance, and overall effectiveness of the tool during honing. Understanding these properties helps to explain how the electroplated layer influences the oilstone's lifespan, particularly when used on different materials. The performance of the electroplated layer is not the focus of this article, so the impact of the electroplating process on the electroplated layer is not elaborated in detail in this article. In addition, composite electroplating is a mature process that has been widely used in the production of grinding tools. Generally speaking, the mechanical and chemical properties of electroplated coatings can meet the processing requirements, ensuring that the grains have sufficient bonding strength and the coating has sufficient chemical stability.
Comments 2.1.2:
2.1.2) How does the increased grain concentration in electroplated oilstones quantitatively impact material removal rates compared to sintered oilstones under identical conditions?
Response 2.1.2:
Under the same type of abrasive particles and grain size, the different preparation processes of electroplated oilstone and sintered oilstone result in significant differences in their grain concentration. The increased concentration of grains in electroplated oilstones leads to a higher number of cutting edges in contact with the material being honed at any given time. This directly translates into a higher material removal rate. The electroplated oilstones allow for more aggressive cutting, as more grains are engaged in the process, reducing the time required to remove material.
Comments 2.1.3:
2.1.3) What are the thermomechanical challenges in the electroplating process, and how might these affect the uniformity or longevity of the grains?
Response 2.1.3:
Thank you for your valuable question. The thermomechanical challenges in the electroplating process—such as thermal expansion mismatches, heat generation, electrolyte composition, cooling stresses, residual stresses, and plating thickness—can significantly impact the uniformity and longevity of the grains on the electroplated oilstone. Non-uniform grain distribution, cracking, delamination, or rapid wear are some of the potential consequences of these challenges. To mitigate these effects, careful control of the plating parameters, including temperature, current density, electrolyte composition, is necessary to ensure that the electroplated layer is both uniform and durable, providing long-term performance during honing.
However, the electroplating process is not the focus of this article, so the impact of electroplating process on the properties of oilstone is not reflected in the text. In addition, electroplating is a mature process for preparing grinding tools. At present, the electroplating process has been optimized for the performance of oilstones, including current density, electrolyte composition, and temperature. The thermodynamic effect is no longer a difficult point in the preparation of electroplated oilstones.
Comments 2.1.4:
2.1.4) What specific parameters (e.g., current density, deposition time) in the electroplating process have the greatest influence on grain exposed height and distribution?
Response 2.1.4:
Thank you for your valuable question. In the electroplating process, several parameters significantly influence the grain exposed height and distribution of the abrasive particles on the substrate. As for grain exposed height, higher current density tends to promote faster deposition of metal, resulting in a thicker bond layer around the abrasive grains, which can reduce their exposed height. Longer deposition times result in a thicker metal layer, which can further encapsulate the abrasive grains. As for grain distribution, high current density can cause non-uniform grain distribution, leading to clustering of the abrasive particles in some regions and sparser coverage in others. A longer deposition time might allow more even grain distribution as the deposition process can more thoroughly cover the surface. However, the preparation process of electroplated oilstone is not the focus of this article, so it is not reflected in the text.
Comments 2.1.5:
2.1.5) How does the initial morphology of the nickel matrix impact the adhesion and stability of the grains during prolonged honing operations?
Response 2.1.5:
Thank you for pointing of this out. The initial morphology of the nickel matrix—including its surface roughness, porosity, composition, microstructure, hardness, and toughness—directly impacts the adhesion and stability of abrasive grains during prolonged honing operations. For example, porous matrix can improve adhesion as it allows for better mechanical bonding between particles and matrix, as pores provide micro anchors for particles. However, excessively high porosity may reduce the overall mechanical strength of the bond and lead to grain loss or peeling during the honing process. After careful consideration, we believe that the current structure and content of the article can clearly convey our research objectives and conclusions, so there is no need to add these additional contents about the initial morphology of the electroplated oilstone nickel matrix.
Comments 2.2:
2.2. Plate Scratch Test
This sub-paragraph could be improved by addressing scaling concerns, introducing real-time monitoring, and linking the test outcomes more directly to practical honing applications.
Response 2.2:
Thank you for your valuable question. The section 2.2 has been modified accordingly based on the comments.
Comments 2.2.1:
2.2.1) How do changes in cutting speed impact the microstructural deformation and subsurface damage in the scratched material?
Response 2.2.1:
Thank you for your valuable question. In summary, higher cutting speeds typically lead to more significant thermal effects, including subsurface damage caused by thermal softening, microcracks, and strain hardening, which can degrade material properties over time. But for honing, low cutting speed and sufficient cooling will not produce significant thermal effects. Moreover, the high exposed of electroplated oilstone abrasive particles will not cause temperature rise due to blockage.
The specific modifications are as follows:
2.2. Plate scratch test
The flat scratch test was conducted on a DMG machining center. During test, the spindle was locked to prevent rotation, and the oilstone fixture was connected to the spindle via the tool holder. The flat oilstone was fixed to the fixture with bolts, applying a certain pressure on the plate, and sliding along the width direction of the oilstone at a certain speed, ultimately leaving scratches on the surface of the plate, as shown in Figure 9. The plates were made of 9Cr18MoV, with their chemical composition detailed in Table 1. The coolant used was Castrol 9554, a water-based emulsion with a concentration of 5%, applied at a pressure was 4 bar. The spindle fed negatively along the Z-axis, which drove the sleeve of the oilstone fixture downward, compressing the spring in the fixture. The spring’s reaction force acted on the piston, transmitting the normal force to the oilstone. The normal force applied to the oilstone was proportional to the spring's deformation. The length and width of the flat oilstone were identical to the actual oilstone, but the trajectory interaction of the oilstone was not considered. A Kistler 9129AA dynamometer was used to measure the normal and tangential forces during the scratch test. The groove topography of the processed surface was measured by an optical profiler. The groove cross-sectional area was further processed to determine the equivalent honing depth. In the flat scratch test, the parameters to be studied included cutting speed, normal force, and overtravel. The specific parameters are listed in Table 2. For plate scratch test, low cutting speed and sufficient cooling will not produce significant thermal effects. Moreover, the high exposure of electroplated oilstone abrasive particles will not cause temperature rise due to blockage. Thus, the influence of cutting speed on the microstructural deformation and subsurface damage can be ignored.
Comments 2.2.2:
2.2.2) What role does the grain exposure height in the oilstones play in determining the depth and uniformity of the grooves during scratching?
Response 2.2.2:
Thank you for your valuable question. The exposed height of the grains in the oilstone plays an important role in determining the depth and uniformity of grooves generated during scratching or honing processes. In order to ensure the stability of honing processing, whether it is sintered oilstone or electroplated oilstone, it needs to be dressed before scratching and honing to remove loosely fixed grains and reduce the height of exposed high grains. The dressed oilstones have better consistency in the exposed height of the grains. Different honing pressures will result in varying numbers and depths of scratches.
Comments 2.2.3:
2.2.3) How does the contact pressure distribution vary across the oilstone surface during the plate scratch test, and what implications does this have for material removal stability?
Response 2.2.3:
The flat oilstone is pressed onto the surface of the plate sample by the reaction force of the spring, and the force acts vertically on the oilstone, as shown in Figure 9. The grains slowly embed into the workpiece, and the directional deformation resistance of the material is balanced with the honing pressure. The honing depth tends to a stable value, indicating the entry into the stable material removal stage.
Comments 2.2.4:
2.2.4) What is the relationship between the groove cross-sectional area and the tribological performance of the machined surface in actual honing applications?
Response 2.2.4:
Thank you for your valuable question. The cross-sectional area of the groove in honing plays a crucial role in determining the frictional properties of the machined surface. A larger groove area can usually increase the retention of lubricant, reduce friction, enhance wear resistance and heat dissipation. However, excessively large grooves can lead to fluid dynamics instability and surface roughness.
Comments 2.2.5:
2.2.5) How do the wear mechanisms of sintered and electroplated oilstones differ when subjected to repeated high-normal-force scratching?
Response 2.2.5:
The wear mechanisms of sintered and electroplated oilstones differ significantly under repeated high-normal-force scratching due to their contrasting structures and bonding methods for abrasive grains.
Sintered oilstones generally wear more gradually due multilayer structure, providing longer tool life under sustained high-normal-force scratching. Electroplated oilstones, on the other hand, are more prone to rapid grain loss due to the limited bond depth and lack of grain replenishment. The choice between the two depends on the application requirements, with sintered oilstones being preferable for durability and electroplated oilstones for precision and short-term high-performance tasks.
More importantly, for electroplated oilstone and sintered oilstone, appropriate normal force should be selected, which can avoid abnormal detachment of grains due to excessive normal force.
Comments 2.3:
2.3. Honing Test
The honing test section could be enriched by including surface morphology analysis, exploring parameter interactions, and addressing oilstone wear. Expanding the applicability of the findings to a broader range of materials and applications would further enhance the relevance and impact of this section.
Response 2.3:
Thank you for your valuable question. The section 2.3 has been modified accordingly based on the comments.
Comments 2.3.1:
2.3.1) How does the relationship between honing pressure and overtravel affect the uniformity of material removal along the sleeve’s depth?
Response 2.3.1:
Thank you for providing such valuable feedback. The main text has already provided corresponding explanations, as follows:
When the honing pressure P remains constant, the normal force Fn acting on the surface of the valve sleeve also remains constant. According to Equation (2), the larger the overtravel, the greater the equivalent honing depth, which causes the honing depth at both ends of the valve sleeve to be greater than that in the middle. Furthermore, under high honing pressure, this phenomenon is more pronounced. Additionally, the contact time between the oilstone and the workpiece varies at different hole depths. Therefore, using the ratio of the total MRV to time to represent the relationship between material removal rate (MRR) and honing pressure is inaccurate. The honing MRR is defined as the MRV per unit time [22, 23], or the MRV per unit time and per unit contact area [24], which does not account for the constantly changing contact state between the oilstone and the workpiece, thus only reflecting the overall material removal capability of the oilstone, rather than accurately representing the material removal situation at different hole depths.
Comments 2.3.2:
2.3.2) What roles do variations in honing speed play in controlling the distribution of material removal across different sections of the valve sleeve?
Response 2.3.2:
Thank you for pointing of this out. The honing speed does not affect the distribution of material removal across different sections of the valve sleeve. The honing speed is the combined speed of rotational speed and reciprocating speed, and only the cross-hatch angle will affect the distribution of material removal across different sections of the valve sleeve. As long as the cross-hatch angle is not excessively large, the impact of cross-hatch angle on removing distribution differences will be weakened with the increase of reciprocating times.
Comments 2.3.3:
2.3.3) How do the contact conditions between the oilstone and the workpiece evolve over multiple honing strokes, and how do they influence MRR stability?
Response 2.3.3:
Thank you for pointing of this out. During multiple honing strokes, the contact conditions between the oilstone and the workpiece undergo an evolution from initial sharp abrasive cutting to later abrasive wear and breakage. With the change of contact area and grain, the stability of MRR is significantly affected, which may lead to fluctuations and instability in MRR. By reasonably controlling factors such as the number of exposed abrasives, lubrication conditions, and oilstone materials, the contact conditions can be optimized to a certain extent to maintain the stability of MRR. All the research and comparisons in this article were conducted under stable cutting conditions of the oilstone, so the influence of oilstone wear can be ignored.
Comments 2.3.4:
2.3.4) What are the wear mechanisms of electroplated oilstones during honing, and how do they compare to sintered oilstones under identical conditions?
Response 2.3.4:
Thank you for pointing of this out. The wear of electroplated oilstone is mainly caused by the detachment and breakage of grains. The detachment of grains is mainly due to the insufficient coating of some grains and the sudden increase in stress. And sintered oilstone, after abrasive wear, new abrasive will gradually be exposed, maintaining continuous cutting ability. However, due to the low exposed of sintered oilstone grains, the oilstone is prone to blockage, resulting in workpiece burns and large oilstone fragments. The research on oilstone wear is not the focus of the study, and the author will conduct research on the mechanism of electroplated oilstone wear in the future.
Comments 2.3.5:
2.3.5) To what extent does the honing-induced microstructure of 9Cr18MoV steel influence its tribological properties and long-term performance in practical applications?
Response 2.3.5:
Thank you for pointing of this out. Generally, by honing the steel surface, higher hardness, refined grain structure, and appropriate residual stress can be obtained, thereby improving the material's wear resistance, fatigue resistance, and corrosion resistance. At present, due to the limitations of oilstone and tools, the honing speed is generally around 1000rpm, and the cutting depth is very low, so the impact on the surface microstructure is not significant.
It is difficult for an article to cover all the research content. The influence of microstructure on valve sleeve performance is not the focus of this study. The author will focus on studying the surface microstructure and properties of honing parts in future research work, mainly focusing on high-speed honing.
Comments 3.1:
- Results and Discussion
3.1. Equivalent Honing Depth
While the findings are scientifically acceptable, the section would benefit from exploring parameter interactions, addressing tool wear, and connecting material removal depth to surface quality and industrial applications.
Response 3.1:
Thank you for your valuable question. The section 3.1 has been modified accordingly based on the comments.
Comments 3.1.1:
3.1.1) How do variations in normal force and overtravel simultaneously influence equivalent honing depth and material removal uniformity?
Response 3.1.1:
The corresponding explanation has been provided in the text regarding this issue, specifically in sections 3.1.2 and 3.1.3.
3.1.2. Normal force
The radial feed of constant pressure honing is closely related to the normal force act-ing on the oilstone. With a honing speed of 5 m/min and zero overtravel, the equivalent honing depth increases as the normal force rises. For the electroplated oilstone, when the normal force increases from 20 to 200 N, the equivalent honing depth increases from 0.0067 to 0.079 μm. In comparison, for the sintered oilstone, as the normal force increases from 20 to 480 N, the equivalent honing depth increases from 0.000047 to 0.0017 μm, (Figure 18). No matter electroplated or sintered oilstone, the equivalent honing depth is basically linear with the normal force, with the slope for electroplated oilstone being significantly steeper than that of sintered oilstone. This is attributed to the higher grain exposed height and concentration of the electroplated oilstone, which results in a larger equivalent honing depth and greater material removal. In addition, the tangential force increases with normal force, but for the same oilstone, the force ratio basically remains unchanged, as shown in Figure 19. Within the normal force variation range of honing, the contact state between the oilstone and the workpiece remains unchanged, leading to a stable force ratio.
3.1.3. Overtravel
Overtravel is a key parameter in honing that directly affects the shape accuracy of the valves. The honing speed is 5 m/min and normal force is 80 N. For constant pressure honing, the MRV at the inlet and outlet of the valve sleeve is affected by the contact time and contact pressure, with the contact pressure being directly impacted by the overtravel. Equation (3) shows that the factors affecting the equivalent cutting depth include the number of scratches, the overtravel, and the cross-sectional area of the material removed by the scratches. When pressure is constant, the number of scratches are slightly reduced as the overtravel increases, while the depth of a single scratch slightly increases. However, the final total cross-sectional area of removed material basically remains essentially unchanged (Figure 20). Under the same normal force, the overall scratch depth of the grain increases with increasing of the contact pressure.
Comments 3.1.2:
3.1.2) What are the underlying mechanisms that cause the equivalent honing depth to stabilize across different honing speeds?
Response 3.1.2:
Different honing speeds can affect the surface microstructure, abrasive contact mode, and thermal effects during cutting, thereby affecting the stability of the equivalent honing depth. A higher honing speed helps to improve the sharpness of abrasive particles and reduce abrasive wear. On the other hand, higher honing speeds can lead to material hardening, resulting in unstable material removal. At present, the honing speed is still only around 1000rpm, which is not enough to cause severe material hardening. Therefore, the current range of honing speeds will not result in unstable material removal.
Comments 3.1.3:
3.1.3) How does the equivalent honing depth correlate with the resulting surface topography and functional properties of the workpiece?
Response 3.1.3:
The equivalent honing depth directly affects the roughness and microstructure of the workpiece surface, thereby affecting the tribological properties, wear resistance, fatigue life, and other functional characteristics. But how the equivalent honing depth specifically affects the functional characteristics of the valve sleeve is not the focus of this article.
Comments 3.1.4:
3.1.4) What role does grain distribution and exposure height in electroplated oilstones play in determining equivalent honing depth under varying conditions?
Response 3.1.4:
Thank you for pointing of this out. This article only studied the effects of 230 mesh CBN grains on material removal ability and surface quality under two oilstone preparation processes, electroplating and sintering. The study of the distribution of electroplated oilstone grains, exposed height, grain concentration, grain size, etc. on the equivalent honing depth will be explored in subsequent research.
Comments 3.1.5:
3.1.5) How do microstructural changes in the workpiece material due to honing affect the equivalent honing depth across different parameters?
Response 3.1.5:
The microstructural changes of the workpiece material during the honing process, such as hardening layer, phase transformation, and microstructural changes, will significantly affect the equivalent honing depth. Specifically:
- The formation of a hardening layer usually increases the cutting difficulty of the material, thereby reducing equivalent honing depth;
- Phase transition and microstructural changes may lead to an increase in material surface hardness or changes in grain structure, thereby affecting cutting efficiency and causing fluctuations in equivalent honing depth;
- Honing parameters such as pressure, speed, and lubrication conditions can also exacerbate or mitigate the effects of microstructural changes, thereby further affecting material removal rates and equivalent honing depths.
The possible influencing factors of microstructure are listed above, and the detailed impact situation needs to be further discussed in subsequent research.
Comments 3.2:
3.2. Material Removal Rate (MRR)
It lacks a discussion of tool wear, surface quality trade-offs, and industrial scalability. Addressing these aspects would make the findings more actionable and relevant to real-world applications.
Response 3.2:
Thank you for your valuable question. The section 3.2 has been modified accordingly based on the comments.
Comments 3.2.1:
3.2.1) How does the interplay between honing pressure and crosshatch angle influence the uniformity of material removal along the workpiece?
Response 3.2.1:
Thank you for pointing of this out. The interaction between honing pressure and cross-hatch angle plays an important role in determining the uniformity of material removal along the workpiece during the honing process. These two parameters, when properly optimized and balanced, help achieve consistent and ideal surface quality, improve material removal rates, and enhance the uniformity of the honing surface. The author has provided explanations and descriptions in the corresponding sections of the text. Specifically, as follows:
3.2.2. Cross-hatch angle
Previous studies have shown that the honing cross-hatch angle affects honing force, oilstone self-sharpening, and contact conditions, thereby significantly impacting honing efficiency.
For sintered oilstones with a honing speed of 25 m/min, as shown in Figure 25, the influence of the cross-hatch angle on diameter change shows that increasing the cross-hatch angle from 16 to 36° results in a decreasing trend in diameter change. Initially, as the cross-hatch angle increases from 16 to 32°, there is a roughly linear downward trend. When the honing pressure is 3 bar, the diameter change decreases by 0.35 μm for every 4° increase. However, when the cross-hatch angle exceeds 32°, the decreasing trend in diameter change accelerates, resulting in a reduction of 1.01 μm for every 4° increase. As the cross-hatch angle increases, the honing trace becomes sparse. As shown in Figure 26, honing trace at a cross-hatch angle of 22.6° (Figure 26(a)) is sparse at a cross-hatch angle of 39.6° (Figure 26(b)). The contact trace between the oilstone and the workpiece be-comes shorter, resulting in a reduction in the MRV and eventually a gradual decline in the diameter change.
As the cross-hatch angle increases from 16 to 36°, the MRR per unit width initially increases and then decreases, as shown in Figure 25. The material removal is stable, with the cross-hatch angle at the inflection point of MRR per unit width occurring around 32°. This inflection point remains consistent across different honing pressures. An increase in the cross-hatch angle results in a decrease in the MRV, but an increase in the cross-hatch angle results in an increase in the reciprocating speed, resulting in a reduction in honing time, as shown in Figure 26. For a honing pressure of 3 bar, when the cross-hatch angle increases from 16 to 32°, the MRR per unit width increases from 0.031 to 0.054 mm3·min-1·mm-1. The reduction of honing time dominates and finally the MRR per unit width increases. When the cross-hatch angle further increases from 32 to 36°, the MRR per unit width decreases from 0.054 to 0.034 mm3·min-1·mm-1. In this case, the decrease in MRR is more significant, outweighing the reduction in honing time, which finally leads to the decrease of MRR per unit width. Thus, to ensure a high MRR, the cross-hatch angle is about 32°.
For electroplated oilstone (Figure 27), the influence of cross-hatch angle on the MRR per unit width follows a similar trend to that of sintered oilstones, with the inflection point also occurring at 32°.
Comments 3.2.2:
3.2.2) What are the dominant mechanisms limiting MRR at higher honing pressures, and how can they be mitigated?
Response 3.2.2:
Thank you for pointing of this out. For the dominant mechanisms limiting MRR at higher honing pressures, the author has provided explanations and descriptions in the corresponding sections of the text. The specific content is reflected from line 441 to 462 of the revised manuscript.
The diameter change of the honed hole indirectly reflects the MRV. Honing was per-formed using a 230# sintered oilstone at a cross-hatch angle of 22.6° and a honing speed of 25.5 m/min. As shown in Figure 22, the effect of honing pressure on the change in hole diameter reveals that when the honing pressure increases from 2 bar to 4.5 bar, the change in hole diameter initially increases and then decreases. The change in hole diameter in-creases from 0.0011mm at 2 bar to 0.0049 mm at 4 bar, due to the increase in normal force caused by the higher honing pressure, which leads to a greater equivalent honing depth, ultimately resulting in a larger MRV. However, when the honing pressure exceeds 4 bar, the MRV begins to decrease. The exposed height of the sintered oilstone grains is relatively low. As the honing pressure increases, the equivalent honing depth also increases, reducing the chip accommodation space. Consequently, the generated chips cannot be expelled promptly and tend to adhere to the oilstone surface. With the increasing number of strokes, chip adhesion becomes more severe, diminishing the oilstone grinding capability, thus leading to a decreasing trend in the change of hole diameter. According to Equation (7), the influence of honing pressure on the MRR per unit width can be derived, as shown in Figure 23, which exhibits the same trend as the hole diameter change. As the honing pressure increases from 2 to 4 bar, the MRR per unit width increases from 0.015 to 0.093 mm³·min⁻¹·mm⁻¹. However, when the honing pressure further increases from 4 to 4.5 bar, the MRR per unit width starts to decrease from 0.093 to 0.078 mm³·min⁻¹·mm⁻¹, which is consistent with the trend and reason for the diameter change. There are two methods to increase the upper limit of honing pressure: improving the oilstone structure to increase more chip space and improving the cooling and lubrication structure to remove chips.
Comments 3.2.3:
3.2.3) How does MRR correlate with the durability and wear resistance of the oilstones under prolonged honing operations?
Response 3.2.3:
Thank you for pointing of this out. The durability and wear resistance of oilstone directly affect its material removal rate (MRR) and stability during honing. Oilstones with good durability can maintain their cutting ability for a longer period of time, thereby ensuring the stability of MRR. Oilstones with good wear resistance can reduce abrasive wear and maintain cutting efficiency. Under long-term honing operations, the wear of the oilstone and the detachment of grains are key factors in reducing MRR. Therefore, the better the durability and wear resistance of the oilstone, the more stable the material removal rate is. Reasonably selecting the type of oilstone and optimizing the honing pressure, speed, and cross-hatch angle according to specific processing conditions can improve the service life of oilstone, thereby increasing material removal rate and stability.
This article uses electroplated oilstone and sintered oilstone, both of which have good wear resistance. The oilstone has been dressed before honing, and all tests were conducted during the stable stage of the oilstone.
Comments 3.2.4:
3.2.4) What role does honing speed variability play in introducing instabilities in material removal, and how can these be minimized?
Response 3.2.4:
Thank you for pointing of this out. The study on how honing speed affects the stability of material removal is detailed in section 3.2.3. Specifically, as follows:
3.2.3. Honing speed
Honing speed is the combined spindle speed and reciprocating speed. Under a certain number of strokes, a faster honing speed results in a shorter honing time. Therefore, honing speed is a key factor affecting machining efficiency.
The honing speed is increased from 10 to 35 m/min by adopting 230# sintered oil-stone and keeping a cross-hatch angle of 24° and a honing pressure of 3 bar, and the diameter change basically keeps unchanged, which is consistent with the results of plate scratch test in Section 3.1.
On one hand, due to the characteristics of the honing process, the oilstone is embedded in the groove, with a certain gap. Compared to the sintered oilstone at a honing speed of 25m/min, excessive honing speed (v=30m/min) can cause the oilstone to experience impacts, resulting in breakage at both ends of the oilstone, as shown in Figure 28, affecting the honing stability, making the error bar of the MRR per unit width bigger and intensifying material removal instability, as shown in Figure 29. The honing axial force at the two honing speeds reveals that the axial force at 25 m/min shows a consistent trend across strokes. However, at the honing speed of 30m/min, the variation trend of axial force in strokes is inconsistent, as depicted in Figure 28. The breakage occurring at both ends of the oilstone alters the contact between the oilstone and the workpiece, leading to fluctuations in axial force that ultimately affect the stability of material removal. Based on the analysis of oilstone state and the characteristics of spindle acceleration and deceleration, excessive honing speed will lead to unstable material removal. For sintered oilstone, it is advisable to set the honing speed at 25 m/min to ensure both high processing efficiency and honing stability.
For electroplated oilstone, the substrate is die steel, which has higher strength and impact resistance than sintered oilstone. The electroplating process ensures a strong bond between the grains and the substrate, making the electroplated oilstone less prone to breaking during honing, unlike sintered oilstone, as shown in Figure 30. Comparing Figure 30(a) and 30(b), there is no significant difference in the morphology of electroplated oilstone before and after honing. Therefore, electroplated oilstones can operate at higher honing speed than sintered oilstones. When the honing speed is 30 m/min, the stability of diameter change can be maintained, as shown in Figure 31. For machining center, the acceleration of spindle in Z direction is fixed. When the reciprocating speed increases from 3 to 7 m/min, the time required to reach the set speed increases from 87 to 110 ms. Higher honing speeds result in a greater Z-direction displacement upon reaching the set speed (as illustrated in Figure 32). The cross-hatch angle is constantly changing, resulting in unstable material removal at the hole inlet and outlet. When the honing speed exceeds 30 m/min, the unstable length exceeds 3.5 mm, exceeding 1/10 of the hole length, negatively affecting shape accuracy and surface morphology consistency. Therefore, by weighing precision and efficiency, the optimal honing speed for electroplated oilstone is determined to be 30 m/min.
Comments 3.2.5:
3.2.5) How do changes in MRR affect the tribological performance of the honed surfaces in practical applications, such as valve sleeve performance?
Response 3.2.5:
Thank you for pointing of this out. High material removal rate usually means strong cutting ability, which may lead to increased surface roughness, especially under high honing pressure or rapid abrasive wear. When the surface is rough, the friction coefficient is often higher because a rough surface can increase the contact area, resulting in more friction. For components such as valve sleeves, this may lead to increased friction losses, heat accumulation, and higher wear rates during operation. How material removal rate affects the performance of valve sleeves is not the focus of this study. On the other hand, if honing is carried out under the same honing parameters, the material removal rate still fluctuates, indicating that the oilstone is blocked or broken, which will directly affect the machining accuracy and surface quality of the valve sleeve, thereby affecting the performance of the valve sleeve.
Comments 3.3:
3.3. Surface Quality
The section could be strengthened by linking surface roughness to functional performance, exploring microstructural changes, and addressing the role of tool wear. These enhancements would provide a more comprehensive understanding of how honing parameters affect surface quality and long-term part performance.
Response 3.3:
Thank you for your valuable question. The section 3.3 has been modified accordingly based on the comments.
Comments 3.3.1:
3.3.1) How does surface roughness influence the tribological performance (e.g., friction and wear) of the honed valve sleeves under operational conditions?
Response 3.3.1:
Thank you for pointing of this out. Surface roughness plays a critical role in the tribological performance of honed valve sleeves, especially under operational conditions involving friction and wear. The surface roughness of the valve sleeve directly affects its interaction with the mating components and can significantly influence performance factors like friction and wear.
Comments 3.3.2:
3.3.2) What microstructural changes occur at the surface and subsurface levels during honing, and how do these correlate with surface quality?
Response 3.3.2:
Thank you for pointing of this out. Firstly, the honing process usually leads to surface hardening, mainly due to the friction and stress effects applied during honing. Secondly, especially under high honing pressure, plastic deformation may occur on the surface and subsurface, which directly affects the surface quality. Lastly, micro defects may occur on the surface of the workpiece due to wear, detachment, and uneven distribution of abrasive particles. The study on the microstructure of honing surfaces is not the focus of this article. Due to the working conditions of low speed and low honing pressure, honing has a relatively small impact on the microstructure of surfaces in oilstone stable cutting stage.
Comments 3.3.3:
3.3.3) How does the wear of oilstones, particularly electroplated ones, affect the evolution of surface roughness during extended honing operations?
Response 3.3.3:
Thank you for pointing of this out. The wear of electroplated oilstones during extended honing operations significantly influences the evolution of surface roughness in the following ways: Initially, the oilstone produces a rougher surface with higher material removal rate. As wear progresses, the oilstone’s abrasive grains lose their sharpness, leading to a decrease in cutting efficiency, which causes the surface roughness to decrease gradually. Eventually, excessive wear can result in an overly smooth or polished surface, with a reduction in the effectiveness of material removal. Therefore, the wear of electroplated oilstones needs to be carefully monitored to maintain a consistent material removal rate and ensure the desired surface finish is achieved throughout the honing process. All studies in this article were conducted during the stable wear stage of oilstones, and the wear of oilstones can be ignored.
Comments 3.3.4:
3.3.4) What are the critical honing pressures at which the balance between material removal rate and surface quality begins to deteriorate significantly?
Response 3.3.4:
Thank you for pointing of this out. For electroplated oilstone, there is no critical honing pressure that significantly deteriorates the surface quality. For sintered oilstone, the critical honing pressure is 3 Bar. The study on the influence of honing pressure on the surface quality of honing has been elaborated in section 3.3. The detailed description is as follows:
The influence of honing pressure and oilstone type on surface roughness is shown in Figure 34. With the increase of honing pressure, the surface roughness of both electro-plated oilstone and sintered oilstone increases. For 230# sintered oilstone, when the honing pressure increases from 2 to 3 bar, the surface roughness remains stable at about 0.066 μm. However, at a honing pressure of 3.5 bar, the surface roughness increases to 0.147 μm. In case of the 230# electroplated oilstone, as the honing pressure increases from 1.8 to 2.55 bar, and the surface roughness escalates from 0.104 to 0.213 μm. According to the processing requirements of the electro-hydraulic servo valve sleeve, the desired surface roughness should be less than 0.1 μm. Although higher processing efficiency can be obtained by honing with 230# electroplated oilstone, the surface roughness of the valve sleeve honed with 230# electroplated oilstone cannot meet the required standards. Additionally, the honing pressure for the 230# sintered oilstone should not exceed 3 bar. To achieve a balance between surface quality and efficiency, it is advisable to allocate the processing allowance strategically. First, use electroplated oilstone for rough honing to remove large amount of material. Then switch to the sintered oilstone for fine honing to ensure the desired surface roughness.
Comments 3.3.5:
3.3.5) How do additional roughness parameters, such as skewness and kurtosis, provide insights into surface quality?
Response 3.3.5:
Thank you for pointing of this out. Skewness and kurtosis are higher-order surface roughness parameters that provide deeper insights into the micro-topography and functional characteristics of a machined or honed surface. While the traditional surface roughness parameters, such as arithmetic mean roughness (Ra), describe the general height profile of a surface. Skewness and kurtosis offer additional information about the distribution and shape of the surface peaks and valleys, which can have important implications for the tribological performance (e.g., friction, wear, lubrication) and functional quality of the surface in practical applications. Subsequently, surface microstructure characterization methods such as skewness and kurtosis will be studied in conjunction with the wear of oilstones in the future.
Comments 3.4:
3.4. High Efficiency Honing of Valve Sleeve
The lack of discussion on tool wear, scalability, and real-world functional performance leaves some critical gaps in the analysis. Addressing these areas would significantly enhance the depth and applicability of the findings.
Response 3.4:
Thank you for your valuable question. The section 3.4 has been modified accordingly based on the comments.
Comments 3.4.1:
3.4.1) How does the wear rate of electroplated oilstones during rough honing influence the efficiency and longevity of the hybrid process?
Response 3.4.1:
Thank you for pointing of this out. The wear rate of electroplated oilstone directly affects its cutting performance, which in turn affects the material removal rate during the rough honing stage. If the oilstone wears out too quickly, its cutting effect will gradually decrease as the abrasive becomes dull or falls off, resulting in a slower material removal rate. At this point, the efficiency of the rough honing process decreases and it takes longer to reach the desired diameter, thereby affecting production efficiency.
Specifically, we believe that although the oilstone wear you mentioned has some value, it will not have a substantial impact on the core argument and results of the article. Our research mainly focuses on the material removal rate and surface quality under the stable cutting state of oilstone, and the part you suggested to add is more related to oilstone wear and life. Although this part is important, it is not the core focus of this article. Under the honing conditions of this article, oilstone wear can be ignored. All studies in this article were conducted during the stable stage of oilstones, and the material can be stably and efficiently removed by the hybrid process. Therefore, we have decided to maintain the existing structure without changing the overall framework of the article.
Of course, we understand and respect your viewpoint. In addition, we are currently conducting relevant research on the wear and lifespan of oilstones, and will also publish related papers in the future.
Comments 3.4.2:
3.4.2) What are the effects of using the hybrid process on the functional performance of valve sleeves, such as tribological behaviour and durability under operational conditions?
Response 3.4.2:
Thank you for pointing of this out. Comments 3.4.2 is similar to A4. The machining quality of valve sleeves affects their service life, especially in terms of shape accuracy and surface quality. Intuitively speaking, using electroplated oilstone and sintered oilstone for composite honing can achieve more stable material removal and better shape accuracy. Secondly, reducing the processing allowance of sintered oilstone can prevent the valve sleeve from being burned due to blockage caused by sintered oilstone. Finally, hybrid honing can achieve the same surface quality as traditional honing, and electroplated oilstone honing can leave some deeper grooves, which is beneficial for oil storage, reducing wear between the valve core and sleeve, and thus extending the life of the valve. This article mainly studies the advantages and disadvantages of surface quality and processing efficiency of electroplated oilstone and sintered oilstone in honing martensitic stainless steel. Therefore, in order to ensure the logical and compact structure of the article, we hope to maintain the current framework. The impact of processing quality on the lifespan of valve sleeves is currently being studied, and corresponding papers will be published in the future to elaborate on this issue in detail.
Comments 3.4.3:
3.4.3) How do process parameters (e.g., overtravel and honing pressure) in the rough and fine honing stages interact to influence overall efficiency and quality?
Response 3.4.3:
Thank you for pointing of this out. The mutual influence of overtravel in rough and fine honing is relatively small, and the function of overtravel is to adjust the shape accuracy of the valve sleeve. rough honing uses electroplated oilstone to improve processing efficiency and control shape accuracy, while fine honing uses sintered oilstone to improve surface quality, ultimately achieving efficient and precise honing of servo valve sleeves.
We apologize for any inconvenience caused by unclear expression. We have optimized the relevant content to ensure that readers can better understand and grasp the relevant content in the article. The specific content is as follows:
The difference between the target diameter and the initial diameter is about 0.014mm. If 230# sintered oilstone is directly used for honing, although the roughness can meet the requirements, the efficiency is low. Conversely, while using the 230# electroplated oilstone enhances efficiency, the roughness fails to meet the required standards. Therefore, to bal-ance quality and efficiency, and improve efficiency as much as possible on the premise of meeting quality, the processing of the electro-hydraulic servo valve sleeve is divided into two steps. The specific allowance distribution and processing parameters are outlined in Table 5. According to previous research, in order to ensure the stability and efficiency of material removal, the cross-hatch angle is 32°. To ensure shape accuracy, the overtravel is 8mm.
Comments 3.4.4:
3.4.4) What adjustments are required in the hybrid process to accommodate materials with significantly different hardness or thermal properties than 9Cr18MoV steel?
Response 3.4.4:
Thank you for providing such valuable feedback. Comments 3.4.4 is similar to A5. Any tool has its own suitable processing object. The hardness, thermal conductivity, and chemical properties of the material will affect its removal. Materials with higher hardness, such as alloy steel and tool steel, usually result in greater cutting forces, which can easily lead to increased tool wear. Therefore, it is necessary to appropriately reduce the honing pressure, especially in the rough honing stage, to avoid excessive wear of the oilstone. To avoid surface roughness or excessive tool wear caused by excessively fast feedrates, the feedrate and speed can be appropriately reduced. Moderately slowing down the speed can also help control honing temperature and avoid the formation of heat affected zones. Materials with lower hardness, such as copper alloys and aluminum alloys, have faster material removal rate. Increasing the honing pressure and feedrate appropriately can improve process efficiency, but it is also important to avoid generating excessive heat that can cause material deformation. Softer materials are prone to surface ripples or frictional overheating during the fine honing process, so finer sintered oilstones can be used to ensure surface quality.
However, this article mainly compares and analyzes the honing performance of electroplated oilstone and sintered oilstone on quenched martensitic stainless steel. Your opinion provides direction for future research. In the future, the author will continue to explore the scope of application of electroplated honing oilstone around the processing object and processing requirements. I am currently conducting research on high-precision honing of copper alloy cylinder block, which will analyze the applicability of electroplated oilstone.
Comments 3.4.5:
3.4.5) What are the potential challenges in scaling the hybrid honing process for mass production, and how might these be mitigated?
Response 3.4.5:
Thank you for providing such valuable feedback. When expanding the hybrid honing process to mass production, potential challenges mainly focus on process consistency, tool wear, thermal management, material differences, and cost control. Firstly, automation and digital control technologies are introduced to use online monitoring systems to track and adjust process parameters in real-time, such as feedrate, honing pressure, and oilstone wear. Sensors are used to monitor temperature, vibration, and tool wear to ensure process stability. Secondly, efficient coolant and cooling systems, such as high-pressure coolant injection, are used to ensure that the workpiece and tool maintain appropriate temperature during the honing process and avoid overheating. Finally, through a comprehensive analysis of production costs, select the most suitable abrasives and tools, and weigh their costs and output benefits. The type and quality of abrasives used can be adjusted according to the needs of different products, optimizing the production process.
Comments 4:
- Conclusions
The section would benefit from more quantitative emphasis, expanded future research directions, and a deeper exploration of real-world implications. Including these elements would elevate the conclusions from a summary of findings to a roadmap for further innovation and practical implementation.
Response 4:
Thank you for your valuable question. The conclusion has been modified accordingly based on the comments.
Comments 4.1:
4.1) What specific innovations in electroplated oilstone preparation could enhance surface quality while maintaining high efficiency?
Response 4.1:
Thank you for providing such valuable feedback. The electroplating process in our research is indeed a mature technology, and the innovation of the article is not reflected in the process itself, but in the specific application of electroplated oilstone and its honing process in this study. Our research focuses more on how to apply this mature technology to the honing field to solve the problem of honing martensitic valve sleeves, thereby demonstrating its superiority and practical value.
Comments 4.2:
4.2) How does the hybrid honing process impact the long-term performance and durability of components in demanding applications (e.g., high-pressure hydraulic systems)?
Response 4.2:
Thank you for providing such valuable feedback. The honing process is suitable for the processing of hydraulic components and can achieve high surface quality and shape accuracy. The cross-hatch microstructure formed on the surface by hybrid honing is beneficial for the storage of hydraulic oil and reduces the friction and wear of hydraulic components. The surface of the workpiece after honing is generally residual compressive stress, which can further improve the wear resistance of hydraulic components.
Comments 4.3:
4.3) What are the critical wear mechanisms of electroplated and sintered oilstones during the hybrid process, and how can they be mitigated?
Response 4.3:
Thank you for providing such valuable feedback. The wear mechanisms of sintered and electroplated oilstones differ significantly under the hybrid honing process due to their contrasting structures and bonding methods for abrasive grains. Sintered oilstones generally wear more gradually due to multilayer and replenishable structure, providing longer tool life. Electroplated oilstones, on the other hand, are more prone to rapid grain loss due to the limited bond depth and lack of grain replenishment. The choice between the two depends on the application requirements, with sintered oilstones being preferable for durability and electroplated oilstones for precision and short-term high-performance tasks. More importantly, for electroplated oilstone and sintered oilstone, appropriate normal force should be selected, which can avoid abnormal detachment of abrasive particles due to excessive normal force.
Comments 4.4:
4.4) How might the findings be adapted to the honing of geometrically complex components or those made of materials with lower machinability?
Response 4.4:
Thank you for providing such valuable feedback. The volume of material removal is actually the integral of material removal rate over time. Based on the material removal rate per unit width defined in this article, the material removal situation at different depths of the valve sleeve can be further obtained. Combining with the actual size and shape of the original hole, the diameter and shape after honing can be obtained. If the honing pressure can be adjusted in real-time, we can control the material removal in the depth direction of the valve sleeve to obtain the desired shape, which will greatly broaden the application scenarios of honing.
Comments 4.5:
4.5) What role does honing-induced residual stress play in the mechanical performance of honed surfaces, and how could this be optimized through parameter adjustments?
Response 4.5:
Thank you for providing valuable suggestions. The residual stress generated during the honing process has a significant impact on the mechanical properties of the workpiece surface. Residual stress can enhance surface fatigue resistance, crack resistance, and wear resistance. We have carefully considered your suggestion regarding the impact of honing on residual stress and have reevaluated the overall structure and research objectives of the article.
We believe that the main objective of this article is to focus on the feasibility of electroplated oilstone in honing martensitic stainless steel, and to explore in depth the differences in material removal rate and surface roughness between electroplated oilstone and sintered oilstone through flat plate scratching and honing tests. Although the influence of honing on residual stress is indeed a direction worthy of further research, this part is not the focus of this study and does not directly affect the main conclusion of the article. Therefore, in order to ensure the logical and compact structure of the article, we hope to maintain the current framework and leave related research on residual stress for further exploration in future work.
Round 2
Reviewer 2 Report
Comments and Suggestions for Authors
The authors answered clearly the reviewer’s concerns.
New interesting applications and future works can be envisioned.